# GL-Disen: Global-Local disentanglement for unsupervised learning of graph-level representations

## Abstract

Graph-level representation learning plays a crucial role in a variety of tasks such as molecular property prediction and community analysis. Currently, several models based on mutual information maximization have shown strong performance on the task of unsupervised graph representation learning. In this paper, instead, we consider a *disentanglement* approach to learn graph-level representations in the unsupervised setting. Our work is the first to study disentanglement learning for graph-level representations. Our key observation is that the formation of many real-world graphs is a complex process with *global* and *local* generative factors. We hypothesize that disentangled representations which capture these global and local generative factors into independent latent units can be highly beneficial. Specifically, for graph-level representation learning, our disentanglement approach can alleviate distraction due to local variations of individual nodes or individual local neighbourhoods. We propose a VAE based learning algorithm to disentangle the global graph-level information, which is common across the entire graph, and local patch-level information, which varies across individual patches (the local subgraphs centered around the nodes). Through extensive experiments and analysis, we show that our method achieves the state-of-the-art performance on the task of unsupervised graph representation learning.

## 1 Introduction

Graph structured data has been very useful in representing a variety of data types including social networks (Newman & Girvan, 2004), protein-protein interactions Krogan et al. (2006), scene graphs (Krishna et al., 2016), customer purchasing patterns (Bhatia et al., 2016) and many more. Graph Neural Networks (GNNs) have recently become the prominent approach for representing graph structured data (Li et al., 2016; Gilmer et al., 2017; Kipf & Welling, 2017; Velickovic et al., 2018; Xu et al., 2019). GNNs are capable of representing graphs in a permutation invariant manner, enabling information propagation among neighbours and mapping graphs to low dimensional spaces.

In this work, we focus on graph-level representation learning. Graph-level representation learning is crucial for tasks like molecular property identification (Duvenaud et al., 2015) and community classification based on the patterns of discussion threads (Yanardag & Vishwanathan, 2015), and they are useful for applications such as drug discovery and recommendation systems. Availability of task specific labels plays a significant role in graph representation learning as much as its role in other domains such as images, text and speech. However, due to many specialized fields which graphs are utilized (e.g., biological sciences, quantum mechanics), collecting labels has become very expensive as it needs expert knowledge (Sun et al., 2020). Therefore, unsupervised learning of graph representation is crucial.

Recent state-of-the-art unsupervised graph representation learning methods (Sun et al., 2020; Hassani & Khasahmadi, 2020) are based on Infomax principle by Linsker (1988). These methods learn the graph representation by maximizing the mutual information between the representation of the entire graph and the representations of individual *patches* of the graph. Here we follow (Velickovic et al., 2019; Sun et al., 2020) and define patches as local subgraphs centered around a node. This

approach allows the graph level representation to capture the globally relevant information from the patch representations (Sun et al., 2020).

**Global-local disentanglement.** We propose a novel approach for graph-level representation learning. Our observation is that many graphs are generated using multiple heterogeneous factors, with different factors providing different information. Specifically, the formation of many real-world graphs is driven by graph-level factors and node/patch-level factors. For example, an online discussion thread can be represented as a graph where nodes represent users who have participated in the discussion thread, and edges represent interaction between the users in the thread (Yanardag & Vishwanathan, 2015). Graph-level representation of such communication graphs can be used to classify the sub-community (e.g. subreddits on Reddit) of that discussion thread. However, the formation of these communication graphs is driven by global graph-level factors (e.g., the topic of the discussion-thread) and local node-level factors (e.g., characteristics of individual users engaging in on-line discussion). The graph is formed with a complex process involving complicated interactions between global graph-level factors and local node/patch-level factors.

It has been discussed in the literature that disentangling these generative factors can benefit many tasks in different domains (Bengio et al., 2013; Ridgeway, 2016). This is because disentanglement enables to separate out explanatory generative factors which cause variations in data and facilitates selection of only those factors which are well suited for the downstream task. Importantly, removing the irrelevant factors from the prediction process increases the robustness of models (Ma et al., 2019).

Based on the above discussion, we hypothesize that graph representation learning that disentangles the graph-level and node/patch-level factors can be useful for many graph analysis tasks. In particular, the disentangled graph-level representation can be powerful for graph-level inference. Therefore, in this work, we propose GL-Disen: a global graph level - local node/patch level disentanglement method for graph level representation learning.

Disentanglement learning is a novel direction for GNNs, and it has not been studied for graph level representations learning. Existing work have only focused on disentangling the factors which forms each neighbourhood, based on supervision from downstream tasks (Ma et al., 2019; Liu et al., 2020; Yang et al., 2020) and disentangling node and edge features in attributed graphs (Guo et al., 2020).

To summarize, our contributions are:

- We propose GL-Disen: a novel global graph-level and local node/patch-level disentangling model. To the best of our knowledge, this is the first work of applying unsupervised disentangled learning for graph level representation learning.
- We conduct extensive experiments to verify that our model learns meaningful disentangled global and local representations for graphs. The disentangled global representation achieves outstanding performance in graph classification.

## 2 RELATED WORK

Here we review the most relevant work on unsupervised graph level representation learning to ours. Reviews of disentangle learning and other unsupervised graph learning methods are in Appendix A.

The most recent family of graph embedding methods are based on contrastive learning. Main idea is to train an encoder model to make it learn the contrast in between a representation which captures the structural and statistic information provided by original data and a negative sample. InfoGraph by Sun et al. (2020) was the first graph level embedding model which utilized contrastive learning and this method was inspired by Infomax principle based Deep Graph Infomax (DGI) (Velickovic et al., 2019). It draws negative samples from other graphs and sum pooling is used as the readout function. Multi-view contrastive (CMV) learning method by Hassani & Khasahmadi (2020) enhances InfoGraph by introducing multi-view based data augmentation mechanism which uses contrastive learning to maximize mutual information among multiple structural views of the input graph. On the other hand, Graph Contrastive Coding (GCC) (Qiu et al., 2020) utilizes contrastive learning for learning universal graph embeddings which can be transferred to multiple downstream tasks. Infomax principle based and contrastive learning based methods have produced the best performance for graph embedding models so far.

## 3 GL-DISEN METHODOLOGY

### 3.1 GRAPH GENERATION PROCESS

We follow the general framework of (Higgins et al., 2017). However, our specific disentanglement method is quite different, as will be discussed. Let $\mathbb{D} = \{\mathbb{G}, G_f, L_f\}$ be the set that consists of graphs and their ground truth generative factors for global and local level. Each graph $G = (V, A)$, contains a set of nodes $V$ and $A$ is the adjacency matrix. $G_f$ and $L_f$ represent two sets of generative factors: $G_f$ contains global factors $\mathbf{g_f} \subset G_f$ common for the entire graph and $\mathbf{l_f} \subset L_f$ represents local factors which can differ from local patch to patch. In our model, $\mathbf{g_f}$ and $\mathbf{l_f}$ are conditionally independent given $G$, where $p(\mathbf{g_f}, \mathbf{l_f}|G) = p(\mathbf{g_f}|G) \cdot p(\mathbf{l_f}|G)$. We assume that the graph $G$ is generated using a true world generator which uses the ground truth generative factors: $p(G|\mathbf{g_f}, \mathbf{l_f}) = Gen(\mathbf{g_f}, \mathbf{l_f})$.

### 3.2 GLOBAL GRAPH LEVEL AND LOCAL PATCH LEVEL DISENTANGLEMENT

Our goal is to develop an unsupervised deep graph generative model which can learn the joint distribution of graph $G$, the set of generative factors $\mathbf{Z}$, using only the samples from $\mathbb{G}$. This should be learnt in a way that the set of latent generative factors can generate the observed graph $G$, such that $p(G|\mathbf{Z}) \approx p(G|\mathbf{g_f}, \mathbf{l_f}) = Gen(\mathbf{g_f}, \mathbf{l_f})$. A suitable approach to fulfill this objective is to maximize the marginal log-likelihood for the observed graph $G$ over the whole distribution of latent factors $\mathbf{Z}$.

$$\max_{\theta} \mathbb{E}_{p_{\theta}(\mathbf{Z})}[\log p_{\theta}(G|\mathbf{Z})] \tag{1}$$

For an observed graph $G$, the inferred posterior probability distribution of the latent factors $\mathbf{Z}$ can be described as $q_{\phi}(\mathbf{Z}|G)$. However, the graph generation process we described in Section 3.1 assumes two independent sets of generative factors representing global and local level information relevant for a graph. Therefore we consider a model where the latent factor set $\mathbf{Z}$ can be divided into two independent latent factor sets as $\mathbf{Z} = (\mathbf{Z}_g, \mathbf{Z}_l)$. $\mathbf{Z}_g$ represents the latent factors which capture the global generative factors of $G$ and $\mathbf{Z}_l$ captures the local counterpart. Therefore we can rewrite our inferred posterior distribution as follows:

$$q_{\phi}(\mathbf{Z}|G) = q_{\phi}(\mathbf{Z}_g, \mathbf{Z}_l|G) = q_{\phi}(\mathbf{Z}_g|G)q_{\phi}(\mathbf{Z}_l|G) \tag{2}$$

We discuss in detail the two posteriors: $q_{\phi}(\mathbf{Z}_g|G)$ and $q_{\phi}(\mathbf{Z}_l|G)$. The graph $G$ consists of $|V|$ number of nodes. In a graph data structure, each node is not isolated. They are connected with its neighbours and propagates information. Therefore, we use the term *patch* to indicate the local neighbourhood centered at each node where the node interacts with. Therefore, $q_{\phi}(\mathbf{Z}_g|G)$ and $q_{\phi}(\mathbf{Z}_l|G)$ are the posterior distributions of all these $|V|$ patches. However, if we consider the global latent posterior, it is common for all $|V|$ patches, as the graph $G$ was originally generated with $\mathbf{g_f}$ common for all $V$. Hence, we propose to use a single latent $\mathbf{z}_g$ to capture the global generative factors common for all patches. In particular, we use $q_{\phi}(\mathbf{z}_g|G)$ to model this single posterior. On the other hand, the factors which contribute to generate each patch can vary significantly. Therefore in this model we assume the local latent factors are independent [1]. Therefore, we update Eq. 2 as:

$$q_{\phi}(\mathbf{Z}|G) = q_{\phi}(\mathbf{z}_g, \mathbf{Z}_l|G) = q_{\phi}(\mathbf{z}_g|G) \prod_{i=1}^{|V|} q_{\phi}(\mathbf{z}_l(i)|G) \tag{3}$$

Here $\mathbf{z}_l(i)$ is the latent factor that captures the local generative factors for a patch centered at node $i$. Now, our objective is to make sure the latent factors sampled from global and local latent posterior distributions can capture the global and local generative factors $\mathbf{g}_f$ and $\mathbf{l}_f$ respectively in a disentangled manner. Note that, we aim to only disentangle global latent factors from local latent factors in this work. This is because, since the intention of the global latent $\mathbf{z}_g$ is to capture all the global factors for graph level representation, entanglement among individual factors in either global latent or local latent is not being focused. Thus, this is different from (Higgins et al., 2017). To

---

[1] In the evaluation we assess the validity of this assumption.

enforce the disentangling nature between these two latent factors, first, we try to match each of them to their respective priors $p(\mathbf{z}_g)$ and $p(\mathbf{z}_l)$ separately. We select unit Gaussians ($\mathcal{N}(0, 1)$) as priors. This leads to following constrained optimization problem (Higgins et al., 2017).

$$
\begin{aligned}
\max_{\theta, \phi} \quad & \mathbb{E}_{G \sim \mathbb{G}} \Big[ \mathbb{E}_{q_\phi(\mathbf{z}_g, \mathbf{Z}_l | G)} [\log p_\theta(G | \mathbf{z}_g, \mathbf{Z}_l)] \Big] \\
\text{s.t.} \quad & KL(q_\phi(\mathbf{z}_g | G) \parallel p(\mathbf{z}_g)) < \epsilon \\
& KL(q_\phi(\mathbf{Z}_l | G) \parallel p(\mathbf{Z}_l)) < \eta
\end{aligned}
\tag{4}
$$

where $\epsilon$ and $\eta$ are strengths of each constraint. Following (Higgins et al., 2017), Eq. 4 can be written to obtain the variational evidence lower bound (ELBO) of a Graph Variational Autoencoder (GVAE) (Kipf & Welling, 2016) (Here we call this as GVAE because our input is a graph) with two separate latent representations with additional coefficients as follows:

$$
\begin{aligned}
\mathcal{F}(\theta, \phi, \alpha, \gamma; G, \mathbf{z}_g, \mathbf{Z}_l) \geq \; & \mathcal{L}(\theta, \phi; G, \mathbf{z}_g, \mathbf{Z}_l, \alpha, \gamma) \\
= \; & \mathbb{E}_{q_\phi(\mathbf{z}_g, \mathbf{Z}_l | G)} [\log p_\theta(G | \mathbf{z}_g, \mathbf{Z}_l)] \\
& - \alpha \, KL(q_\phi(\mathbf{z}_g | G) \parallel p(\mathbf{z}_g)) \\
& - \gamma \, KL(q_\phi(\mathbf{Z}_l | G) \parallel p(\mathbf{Z}_l))
\end{aligned}
\tag{5}
$$

Based on Eq.3 we can expand the KL divergence term $KL(q_\phi(\mathbf{Z}_l | G) \parallel p(\mathbf{Z}_l))$ and rewrite our objective function for a single graph $G$ as:

$$
\begin{aligned}
\mathcal{L}(\theta, \phi; G, \mathbf{z}_g, \mathbf{Z}_l, \alpha, \gamma) = \; & \mathbb{E}_{q_\phi(\mathbf{z}_g, \mathbf{Z}_l | G)} [\log p_\theta(G | \mathbf{z}_g, \mathbf{Z}_l)] \\
& - \alpha \, KL(q_\phi(\mathbf{z}_g | G) | p(\mathbf{z}_g)) \\
& - \gamma \sum_{i=1}^{|V|} KL(q_\phi(\mathbf{z}_l(i) | G) \parallel p(\mathbf{z}_l(i)))
\end{aligned}
\tag{6}
$$

The training process maximizes this lower bound for all the graphs in a minibatch $\mathbb{G}_b$ from the full dataset $\mathbb{G}$:

$$
\mathcal{L}_{\theta, \phi}(\mathbb{G}_b) = \frac{1}{|\mathbb{G}_b|} \sum_{r=1}^{|\mathbb{G}_b|} \mathcal{L}(\theta, \phi; G, \mathbf{z}_g, \mathbf{Z}_l, \alpha, \gamma)
\tag{7}
$$

### 3.3 GL-DISEN ARCHITECTURE IN DETAIL

Figure 1 depicts the proposed GL-Disen model. This is a variation of GVAE where we utilize a $N$-layer GNN as the encoder. $n^{th}$ layer of a GNN can be defined in general as

$$
\mathbf{h}_v^{(n)} = \text{COMBINE}^{(n)} \left( \mathbf{h}_v^{(n-1)}, \text{AGGREGATE}^{(n)} \left( \left\{ \left( \mathbf{h}_v^{(n-1)}, \mathbf{h}_u^{(n-1)}, e_{vu} \right) : u \in \mathcal{N}(v) \right\} \right) \right)
\tag{8}
$$

where $\mathbf{h}_v^{(n)}$ is the feature vector of a patch centered at node $v \in V$ at the $n^{th}$ layer after propagating information from its neighbours $u \in \mathcal{N}(v)$. $\mathbf{e}_{vu}$ is the feature vector of the edge between $u$ and $v$ where $(v, u) \in A$. $\mathbf{h}_v^{(0)}$ is often initialized with node features. We use the term GNN to indicate any network which use layers described in Eq. 8.

Then we utilize two separate sets of GNN layers to produce parameters for posterior distributions for global and local level factors. Let $\mathbf{H}^N = \{\mathbf{h}_v^{(N)}\}_{i=1}^{|V|}$ be the output of GNN after $N$ layers. $\boldsymbol{\mu}_g = \text{GNN}_{\boldsymbol{\mu}_g}(\mathbf{H}^N, A)$ and $log\,\boldsymbol{\sigma}_g = \text{GNN}_{\boldsymbol{\sigma}_g}(\mathbf{H}^N, A)$ is used to generate parameters for the Gaussian posterior distributions for global level latent factors; $q_\phi(\mathbf{z}_g(i) | G) = \mathcal{N}(\boldsymbol{\mu}_g(i), diag(\boldsymbol{\sigma}_g(i))), \; \forall i \in \{1 \ldots |V|\}$. Same way we obtain parameters for the local level latent posterior distributions; $q_\phi(\mathbf{z}_l(i) | G) = \mathcal{N}(\boldsymbol{\mu}_l(i), diag(\boldsymbol{\sigma}_l(i))), \; \forall i \in \{1 \ldots |V|\}$, where

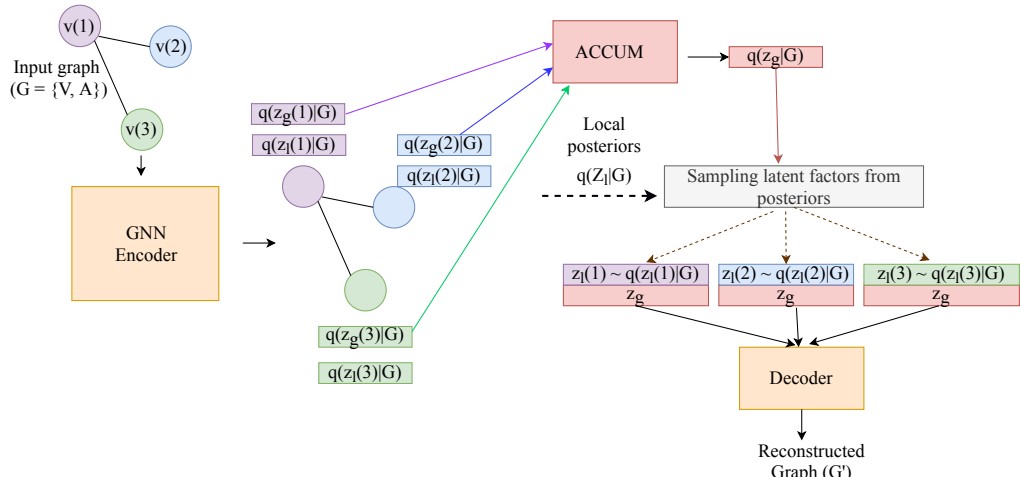

Figure 1: GL-Disen architecture: Given an input graph $G$, we first send it through a GNN to obtain individual global and local latent posterior distributions (Ex: $q(\mathbf{z}_g(1)|G), q(\mathbf{z}_l(1)|G)$) for each patch. Then all individual global posterior distributions $q(\mathbf{z}_g(i)|G)$ are sent through ACCUM process to accumulate into a single posterior $q(\mathbf{z}_g|G)$ for the global latent, using the procedure defined in Eq.9. Sampled local latent factors from their respective local posteriors are combined with the global latent $\mathbf{z}_g$, and this becomes the input to the decoder to reconstruct the graph as per Eq.10. Overall model is trained by maximizing the objective function in Eq.6.

$\boldsymbol{\mu}_l = \text{GNN}_{\boldsymbol{\mu}_l}(\mathbf{H}^N, A)$ and $log\ \boldsymbol{\sigma}_l = \text{GNN}_{\boldsymbol{\sigma}_l}(\mathbf{H}^N, A)$. By utilizing GNN layers for this purpose, GL-Disen allows each patch centered at a node $v$ to disentangle and propagate only the disentangled information to obtain a better posterior distribution for each type of latent.

After obtaining posterior distributions for both global and local latents ($q_\phi(\mathbf{z}_l(i)|G)$ and $q_\phi(\mathbf{z}_l(i)|G), \forall i \in \{1 \dots |V|\}$), we utilize the accumulation step proposed by Bouchacourt et al. (2018): We model the the single posterior distribution $q_\phi(\mathbf{z}_g|G)$ as the product of posteriors $q_\phi(\mathbf{z}_g(i)|G), \forall i \in \{1 \dots |V|\}$. Then, the distribution parameters of $q_\phi(\mathbf{z}_g|G)$ are calculated as follows (Bouchacourt et al., 2018):

$$\boldsymbol{\sigma}_g^{-1} = \sum_{i=1}^{|V|} \boldsymbol{\sigma}_g^{-1}(i), \quad \boldsymbol{\mu}_g^T \boldsymbol{\sigma}_g^{-1} = \sum_{i=1}^{|V|} \boldsymbol{\mu}_g(i)^T \boldsymbol{\sigma}_g^{-1}(i) \tag{9}$$

Then global and local latent generative factors are sampled from their respective posterior distributions ($\mathbf{z}_g \sim q_\phi(\mathbf{z}_g|G)$ and $\mathbf{z}_l(i) \sim q_\phi(\mathbf{z}_l(i)|G), \forall i \in \{1 \dots |V|\}$) and sent through a decoder for reconstructing the graph $G$. Note that the global latent factor $\mathbf{z}_g$ is only sampled once for the entire graph using $q_\phi(\mathbf{z}_g|G)$. We select a simple 2-layer feed-forward neural network with non-linear activations as our decoder $\mathbf{D}$ for the experiments.

A graph reconstruction can be done via both node reconstruction or adjacency matrix reconstruction. Hence we can rewrite the log-likelihood of generating $G$ (first term of Eq. 6) as the combination of generating $V$ and $A$ as follows:

$$\mathbb{E}_{q_\phi(\mathbf{z}_g, \mathbf{Z}_l|G)}[\log p_\theta(G|\mathbf{z}_g, \mathbf{Z}_l)] = \mathbb{E}_{q_\phi(\mathbf{z}_g, \mathbf{Z}_l|V, A)} \sum_{i=1}^{|V|} \log p_\theta(V_i|\mathbf{z}_g, \mathbf{z}_l(i))$$
$$+ \mathbb{E}_{q_\phi(\mathbf{z}_g, \mathbf{Z}_l|V, A)} \sum_{i=1}^{|V|} \sum_{j=1}^{|V|} \log p_\theta(A_{ij}|\mathbf{z}_g, \mathbf{z}_l(i), \mathbf{z}_l(j)) \tag{10}$$

After training GL-Disen in an unsupervised manner, we utilize the latent global $\mathbf{z}_g \sim q_\phi(\mathbf{z}_g|G)$ as the representation which summarizes graph $G$ in downstream tasks such as graph classification.

**Difference between GL-Disen and $\beta$-VAE (Higgins et al., 2017).** While our work is largely inspired by $\beta$-VAE and we apply $\beta$-VAE ideas to graphs, we would like to highlight one key difference

which is critical for this whole work. In particular, $\beta$-VAE baseline cannot discover global factors automatically: $\beta$-VAE discovers independent latent factors, but it is not possible for a baseline $\beta$-VAE to understand if these learned factors are global / non-global in the unsupervised setting. Usually, some manual inspection is performed on the learned latent variables. E.g., for images, one needs to perform traversal of individual latent variables one by one, and observe their effects (e.g., change in azimuth). See Figure 2 of (Higgins et al., 2017). However, as many graphs represent very specialized knowledge, e.g. protein-protein interaction, it is difficult to understand the observed effects and what factors are global/local. Instead, in our work, we add on top of $\beta$-VAE an accumulation step for the GNN encoder outputs of vertices belonging to the same graph (see Fig.1). This forces the emergence of the global factors - common information across all the patches. This mechanism is critical for our idea to extract representation for the whole graph, and we are able to capture global factors without a priori knowledge of the generative factors.

## 4 EXPERIMENTS

Here we discuss our main experiment and analysis results. Additional experiments and analysis are discussed in the Appendix.

Table 1: Mean 10-fold cross validation accuracy on graph classification. Results in **bold** indicate the best accuracy. Underlined results show the second best performances.

| DATASET | MUTAG | PTC-MR | IMDB-BIN | IMDB-MUL | RED-BIN | RED-MUL-5K |
|---|---|---|---|---|---|---|
| node2vec | $72.6 \pm 10.2$ | $58.6 \pm 8.0$ | — | — | — | - |
| sub2vec | $61.1 \pm 15.8$ | $60.0 \pm 6.4$ | $55.3 \pm 1.5$ | $36.7 \pm 0.8$ | $71.5 \pm 0.4$ | $36.7 \pm 0.4$ |
| graph2vec | $83.2 \pm 9.6$ | $60.2 \pm 6.9$ | $71.1 \pm 0.5$ | $50.4 \pm 0.9$ | $75.8 \pm 1.0$ | $47.9 \pm 0.3$ |
| InfoGraph | $89.0 \pm 1.1$ | $61.7 \pm 1.4$ | $73.0 \pm 0.9$ | $49.7 \pm 0.5$ | $82.5 \pm 1.4$ | $53.5 \pm 1.0$ |
| CMV | $89.7 \pm 1.1$ | $62.5 \pm 1.7$ | $74.2 \pm 0.7$ | $51.2 \pm 0.5$ | $84.5 \pm 0.6$ | — |
| GCC | — | — | $72.0$ | $49.4$ | $89.8$ | $53.7$ |
| GVAE(baseline) | $87.7 \pm 0.7$ | $61.2 \pm 1.8$ | $70.7 \pm 0.7$ | $49.3 \pm 0.4$ | $87.1 \pm 0.1$ | $52.8 \pm 0.2$ |
| **GL-Disen**(ours) | **$90.7 \pm 0.6$** | **$67.9 \pm 0.8$** | **$74.7 \pm 0.6$** | **$52.1 \pm 0.4$** | **$90.9 \pm 0.3$** | **$54.9 \pm 0.1$** |

### 4.1 QUANTITATIVE ANALYSIS ON GRAPH CLASSIFICATION

Firstly, we evaluate the effectiveness of learnt disentangled graph level representations from our GL-Disen model on downstream graph classification task.

#### 4.1.1 BASELINES AND RESULTS

We compare our proposed GL-Disen with 6 latest models for unsupervised graph representation learning, which do not employ exhaustive feature selection (Ex: enumerating through paths or subtrees) or hand-crafted filters in this section (Refer Appendix F for a comparison with kernel methods). They only use node features and adjacency matrix on GNNs to propagate information and learn graph representations. We compare with all existing work to the best of our knowledge, which are; node2vec (Grover & Leskovec, 2016), sub2vec (Adhikari et al., 2018), graph2vec (Narayanan et al., 2017), InfoGraph (Sun et al., 2020), CMV (Hassani & Khasahmadi, 2020) and GCC (Qiu et al., 2020). We also use GVAE (Kipf & Welling, 2016) as a baseline to indicate the performance of entangled representations. We use the same evaluation procedure followed by existing work (Yanardag & Vishwanathan, 2015; Sun et al., 2020; Hassani & Khasahmadi, 2020) for a fair comparison. 10-fold cross validation accuracy is used to report the performance and the mean accuracy and standard variation of 5 repeated runs is used as the final result. Complete details of our experiment setup is included at the Appendix E.

Performance comparison of GL-Disen with state-of-the-art unsupervised graph representation learning models are reported in Table 1. We used 6 commonly used datasets; MUTAG, PTC-MR, IMDB-BINARY, IMDB-MULTI, REDDIT-BINARY and REDDIT-MULTI. Appendix D contains complete details. For existing work, we report results from previous papers. Our global graph level and local patch level disentanglement based GL-Disen model has been able to surpass all latest GNN based unsupervised methods. It is important to mention that, although CMV (Hassani & Khasahmadi,

2020) uses additional diffusion matrix to expand the neighbourhood information propagation and GCC pretrains (Qiu et al., 2020) its models with larger datasets, our GL-Disen which only uses adjacency matrix without any pretraining has been able to surpass them. This shows the effectiveness of disentangling and removing non-global factors for global level downstream tasks.

## 4.2 QUALITATIVE ANALYSIS ON GL-DISEN

In order to qualitatively evaluate GL-Disen, we focus on two main aspects. First is, whether GL-Disen actually have the ability to disentangle global and local level factors into independent representations. Second is, whether the inferred global latents correspond to the fixed global generative factors of the graph.

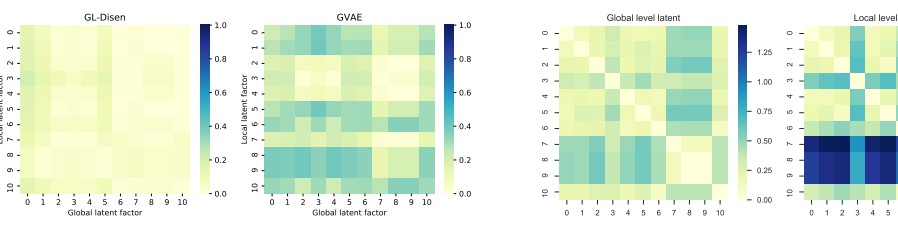

(a) Disentanglment ability of GL-Disen vs GVAE

(b) Patch-wise MAPD for global(left) and local(right) factors

Figure 2: Analysis on GL-Disen: (a) Disentanglement ability of learnt global and local latent factors by GL-Disen via absolute values of correlation compared with GVAE, which does not perform global/local disentanglement. (b) Inter-patch MAPD among global and local latent factors. Lower MAPD for global factors indicates the global factor representations disentangled by GL-Disen is indeed shared among the entire graph, unlike local factors which are local to certain patches.

### 4.2.1 ARE THE LEARNT GLOBAL AND LOCAL LATENT FACTORS BY GL-DISEN DISENTANGLED?

To answer this question, we measure the similarity and dependence between global $\mathbf{z}_g$ and local $\mathbf{z}_l$ latent representations for a given graph $G$. Following Ma et al. (2019), we calculate correlation between disentangled global latent representation and local latent representations for graphs from MUTAG (Kriege & Mutzel, 2012) dataset and visualize in Figure 2(a). Following UDR$_s$ (Duan et al., 2020), we use Spearman's correlation to calculate the similarity/ correlation matrix. As the reference to demonstrate the difference, we use the output correlation for the same graph from non-disentangling model (GVAE) (we divide its single latent representation to halves and consider as $\mathbf{z}_g$ and local $\mathbf{z}_l$). Entry $(i, j)$ of the correlation matrix indicate the absolute correlation value between $\mathbf{z}_g(i)$ and $\mathbf{z}_l(j)$. Note that, only for analysing purposes we sample $\mathbf{z}_g(i) \sim q_\phi(\mathbf{z}_g(i)|G)$.

The diagonal of the correlation matrix shows the correlation between global latent and local latent learnt by GL-Disen for each patch of the graph. We can observe that the correlation values in diagonal is very low, closed to 0.0 for GL-Disen, while correlations between local and global latent variables learned by GVAE have higher values. This shows that the global and local latent representations output by GL-Disen is capable of likely capturing mutually exclusive information showing its disentangling ability.

### 4.2.2 ARE THE INFERRED GLOBAL LATENTS CORRESPOND TO THE GLOBAL GENERATIVE FACTORS OF THE GRAPH?

In order to verify the learned global latents $\mathbf{z}_g$ indeed map to the underlying global factors used for generating graphs, we consider the scenarios when global generative factors are unknown (real-world data) and known (synthetic data).

**Real-world data based experiments.** For real-world graphs, a priori knowledge of the generative factors is usually not available (e.g. molecular graphs). However, following (Higgins et al., 2017), since global generative factors for all the patches of the same graph are fixed, we expect latent variables corresponding to global generative factors to have small variance. There-

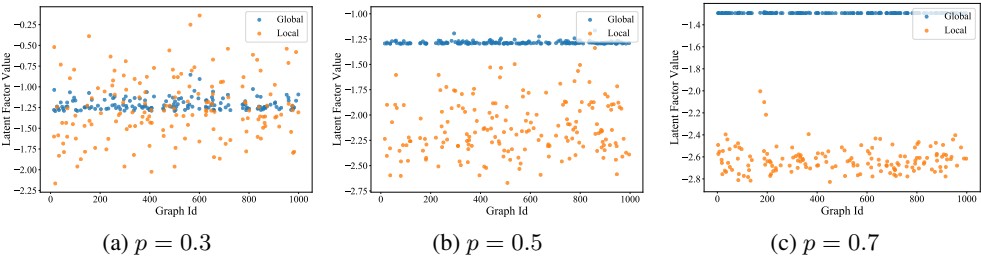

(a) $p = 0.3$      (b) $p = 0.5$      (c) $p = 0.7$

Figure 3: Variation of $\mathbf{z}_g$(Blue) and $\mathbf{z}_g^l$(Orange) of each graph generated with different $p$ values

fore, we check if global latent variables extracted from each patch should be similar to each-other than their local counterparts. Since multiple global generative factors may contribute for the graph generation and the information propagation is different from patch to patch, we cannot assume that the extracted global factors to be exactly the same. But they should be more similar than extracted local latent factors. We evaluate this using the Mean Absolute Pairwise Difference (MAPD) measure used by Higgins et al. (2017) for the disentanglement metric proposed in $\beta-$VAE. MAPD matrix for global latents is calculated as $\text{MAPD}_g(i,j) = \text{MEAN}( \left| \mathbf{z}_g(i) - \mathbf{z}_g(j) \right| )$ and $\text{MAPD}_l(i,j) = \text{MEAN}( \left| \mathbf{z}_l(i) - \mathbf{z}_l(j) \right| )$ used for local latents. We average over all the dimensions of the latent.

Figure 2(b) shows that the produced global level latent representations are more similar to each other for the entire graph compared to local level representations. Following the argument of Higgins et al. (2017), if the representations are disentangled meaningfully, i.e. independent and interpretable, then, there can only be small variances in the inferred latents that correspond to the *fixed* generative factors. Under our model, the global factors for a given graph are fixed, therefore, we examine the variance of the inferred latents $\mathbf{z}_g(i), \forall i \in \{1 \dots |V|\}$ for a given graph (and we show that indeed these $\mathbf{z}_g(i)$ which correspond to the fixed global factors of the given graph has small variance).

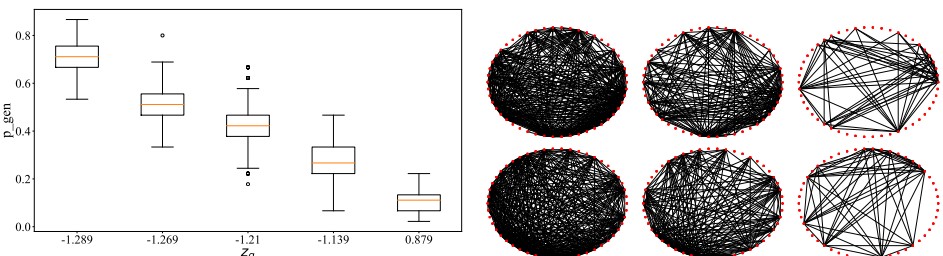

(a) Impact on $p\_gen$ with the increase of $\mathbf{z}_g$      (b) Visualization of generated graphs with the increase of $\mathbf{z}_g$

Figure 4: Impact analysis of global latent representation $\mathbf{z}_g$ on the generation process of GL-Disen and recovering the global generative factor $p$. (a) plots how the distribution of the edge density probability $p\_gen$ (the recovered $p$) changes with the increase of $\mathbf{z}_g$ value. (b) visualizes generated graphs where in each row local latent representation $\mathbf{z}_g$ is fixed and in each column $\mathbf{z}_g$ is fixed. This shows that $\mathbf{z}_g$ has a strong negative correlation with the global generative factor.

**Synthetic graph based experiments** we utilized a synthetic dataset to evaluate this scenario. Synthetic graph dataset was generated using Erdos-Renyi (ER) modelErdos & Renyi (1960). The $\text{ER}(n, p)$ graphs are synthetic graphs with two global generative factors: number of nodes $n$ and a parameter $p \in [0, 1]$ for the synthetic graph to include an edge $(i, j)$ for $1 \leq i < j \leq n$ with probability $p$. There is randomness in each generated graph for a single $p$ value due to many different edge combinations can represent $p$. In our experiments, we focus on parameter $p$, as $n$ is too easy to learn. Therefore, we create a training dataset of $\text{ER}(n, p)$ with fixed $n = 50$ and varying $p$. We generated 4000 graphs in our dataset where 3000 is used for training and the remaining 1000 for testing in which adjacency matrix is the only input for GL-Disen. We use a simple 2 layer GL-Disen

model with hidden dimension size 2 and dimensions of $\mathbf{z}_g$ and $\mathbf{z}_l$ are 1. We analyze two aspects from this experiment.

First is, whether our $\mathbf{z}_g$ corresponds to the fixed global generative factor $p$ while $\mathbf{z}_l$ is not. To achieve this, we analyzed how $\mathbf{z}_g$ and $\mathbf{z}_l$ vary with fixed $p$. Figure 3 shows scatter plots for 3 different values of $p$. From the testing set, we selected the set of graphs generated using the given $p$ and we sampled $\mathbf{z}_g$ as the global latent representations for each graph. Then we calculated an accumulated single local latent representation ($\mathbf{z}_g^l$) for each graph using $\mathbf{z}_l$ on Eq.9 to observe local latent factor variation with $p$. We plotted the values of $\mathbf{z}_g$ and $\mathbf{z}_g^l$ for each graph for the given $p$ in Figure 3. We can see that while values of $\mathbf{z}_g$ are scattered within a very small range (almost similar to a constant) when the global generative factor $p$ is fixed, $\mathbf{z}_g^l$ has varied a lot. This shows us that indeed the global latent representation $\mathbf{z}_g$ in GL-Disen has been able to map accurately to the global generative factor while the local latent representations have not. Next, we calculated the Pearson r correlation as well as co-variance to understand how $\mathbf{z}_g$ changes with $p$. We obtained that $\mathbf{z}_g$ has a very strong negative correlation with $p$ of value 0.93 and the co-variance is 0.248. We further confirm this from the generative process of GL-Disen in Fig. 4 & 10 where we qualitatively showcase how global and local latent factors effect the graph generation process of GL-Disen.

### 4.3 CAN NODE LEVEL TASKS BENEFIT FROM GL-DISEN TOO?

In this section we evaluate the impact of disentanglement on node level tasks to see whether node level tasks only benefit from local information or not.

Table 2: Mean test set accuracy on node classification. Results in **bold** indicate the best accuracy for each dataset. Value in brackets in the last row indicates the $\eta$ value which gave the best performance.

| DATASET | CORA | CITESEER | PUBMED |
|---|---|---|---|
| GVAE (Kipf & Welling, 2016) | $77.8 \pm 0.4$ | $60.3 \pm 0.6$ | $61.5 \pm 0.4$ |
| Disentangled Graph only | $14.4 \pm 8.7$ | $15.7 \pm 4.7$ | $33.7 \pm 10.9$ |
| Disentangled Node only | $75.5 \pm 0.6$ | $62.2 \pm 0.4$ | $72.4 \pm 0.3$ |
| Combined | $\mathbf{78.8 \pm 0.4}$ ($\eta = 0.4$) | $\mathbf{67.8 \pm 0.2}$ ($\eta = 0.45$) | $\mathbf{77.4 \pm 0.4}$ ($\eta = 0.9$) |

To find this out, we updated our GL-Disen decoder $\mathbf{D}$, with a gating mechanism, where our gate value $\eta$ determine the contribution from local patch representation $\mathbf{z}_l(i)$. combined input of $\hat{\mathbf{z}}_l(i) = \eta \, \mathbf{z}_l(i) + (1 - \eta) \, \mathbf{z}_g$ is used as input to decoder $\mathbf{D}$ during training time and also used as the input to our node level downstream-task node classification. GVAE (Kipf & Welling, 2016) which produces entangled representations is out baseline. We use the citation network benckmark (Cora, CiteSeer and PubMed) Yang et al. (2016) for our experiments. Table 2 contains the performance of GL-Disen on node-level task when only graph level information is utilized ("Graph only" $\rightarrow \eta = 0$), when only patch information is used ("Node only" $\rightarrow \eta = 1$) and the best performance obtained by combining both (Appendix I contains detailed discussion). We can observe that for all datasets, combining both information has been beneficial over using only patch level information. Donnat et al. (2018) has stated that identifying distant nodes with similar neighbourhood structures is a strong fact for node level task performance and we believe our combined method increases the performance due to the fact that $\mathbf{z}_g$ (the disentangled global representation) has been able to capture those kind of similarities across the entire graph compared to "Disentangled node only" model which solely relies on its local neighbourhood similarities.

## 5 CONCLUSIONS

We introduced a disentanglement learning based approach for unsupervised graph level representation learning. Our assumption was that disentangling global level information shared among the entire graph, from local level information which are unique for patches, is beneficial for graph level tasks as it removes irrelevant information which can act as noise. We proposed VAE based GL-Disen for this purpose. From both quantitative and qualitative empirical results, we showcased the validity of our assumption and the effectiveness of our model. We achieve new state-of-the-art performance for unsupervised graph representation learning task.

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

# A    RELATED WORK

## A.1    UNSUPERVISED GRAPH LEVEL REPRESENTATION LEARNING

**Kernel Methods**    Graph kernels are widely used for representing graph structured data over decades. Main idea of graph kernels is to first find out best sub-structures which the graphs can be divided into and then enumerate and count the occurrences of these sub-structures to represent them as a high dimentional feature vector. Most common substructures are walks (Gärtner et al., 2003), shortest paths (Borgwardt & Kriegel, 2005), subtrees (Shervashidze et al., 2011), or graphlets (Shervashidze et al., 2009). Then the graph kernels are defined to calculate pairwise substructure similarity between two given graphs. Earlier graph kernels decoupled the process for kernel based data representations and task based model training into two parts. Therefore unlike GNNs, kernels can neither produce task dependent features, nor can train end to end. However kernels have more expressive power and regularize properly than GNNs. More recent kernel based models like GCKN (Chen et al., 2020) have tried to combine the best of both kernal and GNN worlds by extending convolution kernel networks. However, still these kernel based methods rely on enumeration of the substructure occurrences in graphs, where they gain better expressive power at the cost of efficiency. Kernel based methods which use walk kernels are the most similar to GNN as GNN also uses walks for information propagation. Although doesn't belong to the kernel category, the recent work by Bai et al. (2019) also follows a pair-wise graph similarity caluculation mechanism based on proximity calculations using Graph Edit Distance (Sanfeliu & Fu, 1983) for unsupervised graph representation learning.

**Other Methods**    Another set of methods were proposed inspired by the word2vec skip-gram model from natural language processing to encode neighbourhood information to a vectorized representation and consider it as the graph embedding. First model was node2vec (Grover & Leskovec, 2016) and sub2vec (Adhikari et al., 2018) uses random walks to identify each node's neighbourhood and encode that to a latent vector to represent each node or sub-graph respectively. graph2vec (Narayanan et al., 2017) uses Weisfeiler-Lehman kernel (Shervashidze et al., 2011) to calculate non-linear substructures opposed to linear sub-structures from sub2vec to vectorize full graphs. However these methods are completely dependent of neighbourhood information and unable to utilize node features.

## A.2    DISENTANGLED REPRESENTATION LEARNING

Autoencoder (Baldi & Hornik, 1989; Hinton & Zemel, 1993) based approaches has proven to be one of the most effective methods in representation leaning (Bengio et al., 2013) which provides compact and meaningful representations without any supervision. These learnt latent representations can be utilized for downstream tasks such as classification or clustering. Variational Autoencoders (Kingma & Welling, 2014) further enhance these capabilities by supporting variational inference. Disentangled learning (Desjardins et al., 2012) is focused on learning the model to assign different factors used for object composition to different dimension of the latent representation. VAEs were proven to be capable of disentangling the latent features after regularizing its objective function (Higgins et al., 2017; Burgess et al., 2018). Later more variations of VAE (Chen et al., 2018; Kim & Mnih, 2018; Kumar et al., 2018) have been proposed for better disentangled learning.

Another line of disentangled learning was proposed based on grouped observations by Bouchacourt et al. (2018) which makes the models learn the semantics of provided grouping. These methods aim at disentangling content (group) and style (individual element) level information where the content is common for the entire group while style is independent for each element in a group. Group level acts as weak supervision where the disentanglement is only learnt based on the fact that all element belong to the same group without actually knowing the group label. They propose a multi-layer VAE (ML-VAE) to achieve this and they show the effectiveness of their method based on its generalizability to unseen groups and controllability over latent space.

Inspired by this work, we propose to apply natural grouping among nodes (as they belong to the same graph via edges) to disentangle global graph level factors from local factors.

### A.3 Disentangled representation learning for GNN

In this section, we discuss disentangled representation learning for GNN. We refer the reader to the Appendix for a review of general disentangled representation learning.

Disentangled learning is a very novel direction for graph representation learning domain as only very few work are currently available. DisenGCN (Ma et al., 2019) was the first method proposed to disentangle the node representation which dynamically extracts each factor caused for each neighbour to form an edge with current node. IPGDN (Liu et al., 2020) proposes another disentangled node representation which focuses not only on disentangling factors which relates the node with its neighbours, but also on making those factors as independent as possible. Both of these methods are guided by the supervision of downstream node classification tasks. FactorGCN (Yang et al., 2020) is another supervised disentangling model which uses a factoring mechanism at the input level to disentangle input features which are to propagate among neighbour nodes to separate factors and form them as separate graphs. Then they use a discriminator to make the factors as independent as possible. Then each of these graphs are separately sent through a GNN and aggregated together. task supervision. NED-VAE (Guo et al., 2020) is an unsupervised disentangling model which disentangles node and edge features from attributed graphs.

Compared to these methods, we follow a different level of disentanglement in an unsupervised manner. Our GL-Disen focuses on disentangling global level factors common for the entire graph from local level factors specific to patches. We do not aim at disentangling individual factors in each level. All existing disentangle methods including DisenGCN(Ma et al., 2019), IPGDN(Liu et al., 2020), FactorGCN(Yang et al., 2020) and NED-VAE with all its variations (Guo et al., 2020) can be identified as disentangling individual factors in local level as they either factorize the neighbourhoods or disentangle node and edge features. They do not aim at disentangling and separating out factors common for the entire graph.

## B   Pseudo code of our proposed GL-Disen

---
**Algorithm 1** GL-Disen Algorithm
---
1: **for** Each epoch **do**
2:     Sample a minibatch of graphs $\mathbb{G}_b$
3:     **for** $G \in \mathbb{G}_b$ **do**
4:         **procedure** Encoder($G$)
5:             Encode $G \in (V, A)$ into $q(\mathbf{z}_g(i)|G; \phi)$, $q(\mathbf{z}_l(i)|G; \phi)$, $\forall i \in \{1 \dots |V|\}$
6:         **end procedure**
7:         Accumulate $q(\mathbf{z}_g(i)|G; \phi)$, $\forall i \in \{1 \dots |V|\}$
8:         Obtain a single global level latent posterior $q_\phi(\mathbf{z}_g|G)$                          ▷ Eq. 9
9:         **procedure** Decoder($q_\phi(\mathbf{z}_g|G), q(\mathbf{z}_l(i)|G; \phi)$, $\forall i \in \{1 \dots |V|\}$)
10:            Sample a single global latent factor $\mathbf{z}_g \sim q_\phi(\mathbf{z}_g|G)$ for the entire $G$
11:            Sample individual local latent factors for each substructure $i$
12:            **for** $i \in \{1 \dots |V|$ **do**
13:                $\mathbf{z}_l(i) \sim q_\phi(\mathbf{z}_l(i)|G)$
14:            **end for**
15:            Decode $\mathbf{z}_g$ and $\{\mathbf{z}_l(i)\}_{i=1}^{|V|}$ to reconstruct the graph $G$ using $p(G|\mathbf{z}_g, \{\mathbf{z}_l(i)\}_{i=1}^{|V|})$
16:        **end procedure**
17:    **end for**
18:    Update $\theta, \phi$ by taking a gradient step of $\mathcal{L}_{\theta,\phi}(\mathbb{G}_b), \nabla_{\theta,\phi}\mathcal{L}(\mathbb{G}_b, \theta, \phi)$          ▷ Eq. 7
19: **end for**
---

## C   Model Complexity Analysis

We like to discuss the time and space complexity of GL-Disen compared to our baseline GVAE(Kipf & Welling, 2016). Most of the computation complexity comes from the GNN encoder (Eq.8) where the time and space complexity is $O(V^2)$ for a single GNN layer with $V$ number of nodes in the graph and for GNN with $N$ layers, it becomes $O(V^2N)$. Only difference between GL-Disen encoder and

GVAE encoder is that due to disentangling, and GL-Disen encoder requires output of two different parameter sets for global and local posterior distributions instead of one as in baseline GVAE. Therefore we need an additional 2 GNN layers. Since it is a constant addition, the overall complexity stays at $O(V^2 N)$ scale. The decoder complexity for both GVAE and GL-Disen is $O(V^2)$ with adjacency reconstruction being the dominant component (Eq.10). The two additional steps GL-Disen have for the disentanglement are as follows (in between encoder and decoder): (A) Accumulating using Eq.9 and (B) combining global and local samples to feed to the decoder. Both step (A) and (B) are linear operations during both training and inference with the complexity of $O(V)$ in both time and space. Compared to the high complexity of the GNN encoder and decoder common for both GVAE and ours, this linear increment to disentanglement is not significant.

## D  EVALUATION DATASETS

We select six commonly used graph classification benchmark datasets as follows: MUTAG (Kriege & Mutzel, 2012) dataset contains mutagenic aromatic and heteroaromatic nitro compounds while PTC dataset (Kriege & Mutzel, 2012) consists of chemical compounds reported for carcinogenicity of rats. Apart from these bioinformatics datasets, next we evaluate on four social network datasets (Yanardag & Vishwanathan, 2015) namely IMDB-BINARY, IMDB-MULTI, REDDIT-BINARY and REDDIT-MULTI-5K. IMDB datasets contain information about movies where the nodes are actors/actresses and they are connected by edges if they have acted in the same movie. IMDB-BINARY contains two genres of movies: Action and Romance. Multi-class version of IMDB dataset contains movies from Comedy, Romance and Sci-Fi genres. Reddit datasets were created using threads in different subreddits. Nodes in each graph are users who responded to that particular thread and edges are formed when one user respond to another user's comment. REDDIT-BINARY dataset labels each graph as question/answer-based community or a discussion-based community and REDDIT-MULTI-5K labels graphs into 5 labels according to their subreddit namely, *worldnews, videos, AdviceAnimals, aww* and *mildlyinteresting*. Table 1 contains statistics of all these datasets. Statistics of the benchmark datasets are in Table 3.

Table 3: Dataset statistics used for graph level tasks

| DATASET | MUTAG | PTC-MR | IMDB-BIN | IMDB-MUL | RED-BIN | RED-MUL-5K |
|---|---|---|---|---|---|---|
| # Graphs | 188 | 344 | 1000 | 1500 | 2000 | 4999 |
| Avg. Nodes | 17.93 | 14.29 | 19.77 | 13.0 | 429.63 | 508.52 |
| Avg. Edges | 19.79 | 14.69 | 96.53 | 65.94 | 497.75 | 594.87 |
| # Classes | 2 | 2 | 2 | 3 | 2 | 5 |

For node classification task, we used the citation network datasets; Cora, CiteSeer and PubMed) Yang et al. (2016). Statistics of the datasets are in Table 4.

Table 4: Dataset statistics used for node level tasks

| DATASET | CORA | CITESEER | PUBMED |
|---|---|---|---|
| # Graphs | 1 | 1 | 1 |
| # Features | 1433 | 3703 | 500 |
| # Nodes | 2485 | 2110 | 19717 |
| # Edges | 5069 | 3668 | 44324 |
| # Classes | 7 | 6 | 3 |

## E  EXPERIMENT SETUP

We use the same evaluation procedure followed by existing work (Yanardag & Vishwanathan, 2015; Sun et al., 2020; Hassani & Khasahmadi, 2020) for a fair comparison. 10-fold cross validation accuracy is used to report the performance and the mean accuracy and standard variation of 5 repeated runs is used as the final result. Mean Squared Error and Binary Cross Entropy losses are used to

calculate node and adjacency matrix reconstruction errors. After GL-Disen model is trained in un-supervised manner, the classification accuracies are calculated using LIBSVM (Chang & Lin, 2011) which the C parameter was selected from $\{1, 10^1, 10^2, 10^3, 10^4\}$ using cross-validation from the training folds of data.

Following InfoGraph (Sun et al., 2020), Graph Isomorphism Network (GIN) (Xu et al., 2019) is used as the GNN encoder in GL-Disen. We initialize GL-Disen model using Xavier initialization (Glorot & Bengio, 2010) and train the model using Adam optimizer (Kingma & Ba, 2015). Initial learning rate, number of GNN layers and batch size were set to 0.001, 2 and 128 respectively. Number of training epochs and hidden dimension were chosen from $\{20, 30, 40, 50\}, \{32, 128, 512\}$. Refer to the Appendix for the selected hyper-parameter sets for each dataset.

For node classification task, we follow DGI (Velickovic et al., 2019) and report the mean classification accuracy with standard deviation on the test nodes after 50 runs of training followed by a linear model. We use a single layer GCN as the encoder and the hidden dimensions are 512.

Model implementations and data loading were done using Pytorch (Paszke et al., 2017) and Pytorch Geometric (Fey & Lenssen, 2019) for all our experiments.

## F  COMPARISON OF KERNEL BASED MODELS WITH GL-DISEN FOR GRAPH CLASSIFICATION

In this section we compare the performance of our GL-Disen with another line of unsupervised graph representation learning; kernel based methods. We use 7 different models of kernel methods for this; Random Walk (RW) (Gärtner et al., 2003), Shortest Path (SP) (Borgwardt & Kriegel, 2005), Graphlet Kernel (GK) (Shervashidze et al., 2009), Weisfeiler-Lehman Subtree kernel (WL) (Shervashidze et al., 2011), Deep Graph Kernels (DGK) (Yanardag & Vishwanathan, 2015), Multi-Scale Laplacian (MLG) (Kondor & Pan, 2016) and most recent Graph Convolutional Kernet Network (GCKN) (Chen et al., 2020). We only compare with GCKN walk kernel method as it is the closest feature aggregation for GNNs.

Table 5: Mean 10-fold cross validation accuracy comparison on graph classification of Kernel methods with GL-Disen. Results in **bold** indicate the best accuracy for each category. Underlined results show the second best performances.

| DATASET | MUTAG | PTC-MR | IMDB-BIN | IMDB-MUL | RED-BIN | RED-MUL-5K |
|---|---|---|---|---|---|---|
| Kernel Methods | | | | | | |
| RW | $83.7 \pm 1.5$ | $57.9 \pm 1.3$ | $50.7 \pm 0.3$ | $34.7 \pm 0.2$ | $-$ | $-$ |
| SP | $85.2 \pm 2.4$ | $58.2 \pm 2.4$ | $55.6 \pm 0.2$ | $38.0 \pm 0.3$ | $64.1 \pm 0.1$ | $39.6 \pm 0.2$ |
| GK | $81.7 \pm 2.1$ | $57.3 \pm 1.4$ | $65.9 \pm 1.0$ | $43.9 \pm 0.4$ | $77.3 \pm 0.2$ | $41.0 \pm 0.2$ |
| WL | $80.7 \pm 3.0$ | $58.0 \pm 0.5$ | $72.3 \pm 3.4$ | $47.0 \pm 0.5$ | $68.8 \pm 0.4$ | **$46.1 \pm 0.2$** |
| DGK | $87.4 \pm 2.7$ | $60.1 \pm 2.6$ | $67.0 \pm 0.6$ | $44.6 \pm 0.5$ | **$78.0 \pm 0.4$** | $41.3 \pm 0.2$ |
| MLG | $87.9 \pm 1.6$ | $63.3 \pm 1.5$ | $66.6 \pm 0.3$ | $41.2 \pm 0.0$ | $-$ | $-$ |
| GCKN-walk | **$92.8 \pm 6.1$** | **$65.9 \pm 2.0$** | **$75.9 \pm 3.7$** | **$53.4 \pm 4.7$** | $-$ | $-$ |
| Ours | | | | | | |
| GVAE(baseline) | $87.7 \pm 0.7$ | $61.2 \pm 1.8$ | $70.7 \pm 0.7$ | $49.3 \pm 0.4$ | $87.1 \pm 0.1$ | $52.8 \pm 0.2$ |
| **GL-Disen**(ours) | **$90.7 \pm 0.6$** | **$67.9 \pm 0.8$** | **$74.7 \pm 0.6$** | **$52.1 \pm 0.4$** | **$90.9 \pm 0.3$** | **$54.9 \pm 0.1$** |

GCKN (Chen et al., 2020) reports superior performance to ours in 3 datasets although our results are very comparable to them. One of the major aspect of kernel methods is they use manual processes (graph traversals like depth first search) to find all possible paths for substructures like random walks, trees or graphlets. Then they compare all those pairs of paths in each pair of graphs to calculate kernel values to find similarities. This is a very expensive operation. However for small graphs this gives better results as it covers all possible neighbourhoods. However as the GCKN (Chen et al., 2020) mentions, when there are very large dense graphs, they are unable to extend this method. This can be a reason that kernel based methods do not evaluate on denser datasets like Reddit. On the other hand, GNNs achieve efficiency by eliminating from manual path and graph to graph pairwise comparison and reducing neighborhoods for only random walks. However even with

limited neighbourhood, we could see that GNNs specially with our disentanglement mechanism have been able to achieve almost similar performance.

## G   ADDITIONAL ANALYSIS TO PROVIDE MORE EVIDENCE THAT $\mathbf{z}_g$ IS ESSENTIAL TO CAPTURES GLOBAL LEVEL INFORMATION

Table 1 shows that GL-Disen is able to surpass all existing GNN based unsupervised methods for graph classification. However, it is important to compare the performance of disentangled global representation against disentangled local representations as well as entangled representations to verify that the most crucial information for downstream global level tasks (Ex: graph classification) are indeed captured by global latent representation $\mathbf{z}_g$.

For this, the first experiment we conducted was to separately evaluate how the downstream graph classification accuracy gets effected when using disentangled latent global representations, disentangled latent local representations and entangled latent representations. We used the same configurations as the original GL-Disen where $\mathbf{z}_g$ was sent to the SVM. We use the same accumulation function mention in Eq.9 to obtain the single posterior distribution and sample a representation for both disentangled latent local and entangled latent before feeding them to SVM. For obtaining entangled representations, we use a vanilla GVAE.

Next, we examine whether local factors are completely irrelevant for global level tasks. To analyse this, we feed a linear combination of latent global representation and latent local representation to the SVM classifier. This experiment was conducted using the best pre-trained models according to Section E. We set a gating hyper-parameter $\lambda$ to determine the contribution comes from latent local representation. We use the following function to obtain the final graph representation $\mathbf{z}_g$ to be sent to the SVM. We use the same accumulation function mention in Eq.9 for obtaining a single posterior distribution to sample $\mathbf{z}_l$ from.

$$\hat{\mathbf{z}}_g = \lambda\,\mathbf{z}_l + (1 - \lambda)\,\mathbf{z}_g \tag{11}$$

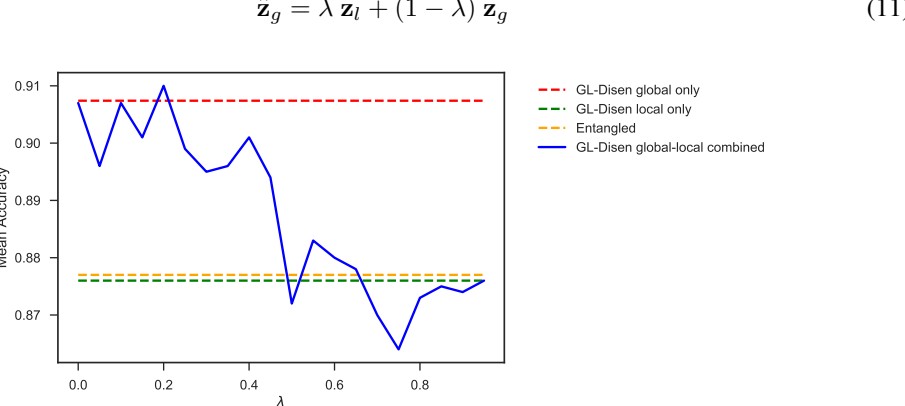

Figure 5: Comparison of the impact of disentangled local only, disentangled global-local combined and entangled (GVAE) latent representations for graph classification performance against disentangled global only $\mathbf{z}_g$ representation. (best viewed in colour)

Figure 5 visualizes the impact of local latent factors $\mathbf{z}_l$ and entangled latent factors (latent representation from vanilla GVAE) against global latent factors $\mathbf{z}_g$ for graph classification task. Each dotted line indicates the accuracy value obtained when each of these latent representations are individually sent to SVM for graph classification. We can observe that global latent representation has achieved the best individual performance surpassing local only and entangled. GL-Disen local only performance has been very close to entangled performance obtained from GVAE. We believe this is because, although global information are included in GVAE's latent representation (in an entangled manner), each observation is treated in an IID manner. Hence those global information might be getting suppressed in order to give space for latent factors which enhances reconstruction ability of the model. Solid blue line shows how the graph classification accuracy changes with increasing $\lambda$. We observe when the $\lambda$ is increasing (more contribution coming from local latent representation),

the classification performance drops. Therefore we can conclude that the global latent $\mathbf{z}_g$ achieves the best performance. And there is no gain in incorporating disentangled local latent $\mathbf{z}_l$ except some random variations.

## H   HOW CAN THIS METHOD PROVE THAT EACH FACTOR IS NECESSARY FOR THE GENERATIVE PROCESS?

In what follows, we discuss how we validate that our learned local and global latent variables carry critical information for the generative process.

As we discussed, our main focus of GL-Disen is to separate out the group of generative factors as local and global, i.e. global/local disentanglement, and that is sufficient for our task. There is no need for us to explicitly separate each one of global and local factors from those sets individually.

To evaluate the necessity of our local/global latent variables, first we calculated the node feature reconstruction error for MUTAG dataset and obtained the following results. MSE when both global and local factors fed to the decoder is 0.03256 and it increases to 0.03654 when global factors are removed (local only). When only global factors are fed to the decoder (global only), the error further increases to 0.08329. From these errors, we can observe that local latent factors have the largest impact on the generation of the node features. This is expected as global factors are common for all the patches for a given graph. Therefore, to reconstruct the node features (which differ from node to node), local factors are crucial. However, we can observe from the difference of full model and local only errors, that our model does not ignore the global factors during node feature generation. Hence showing it is also necessary.

Next, we show that our learned global latent variables carry critical graph level information in the generative process. We refer to Sec. 4.2.2 - Synthetic graph based experiments and Fig. 4 on the updated manuscript. In Fig 4(b), we show generated sample graphs using disentangled global and local factors. In each row of Fig 4(b), the local latent factors are fixed and in each column the global factors are fixed. When we consider a single row, we could observe that, the edge density of the graph changes with the change of global factors. Although 2 rows have two structurally different graphs (nodes have different neighbourhoods), the global factor has been able to change the edge density of those 2 in a similar manner. If only local factors are necessary, then every graph in the same row should look alike. This shows that graph level generative information is captured by global latent variables. Therefore the global latent variables are necessary for the generative process.

Further evidence that our learned global latent variables carry critical graph level information comes from the evaluation on graph classification task. In Appendix G, we evaluate the impact of different combinations of global/local latent (Eq. 11) on graph level task performance. We observe that using only learned global latent variables ($\lambda = 0$) achieves the best performance in graph level classification. On the other hand, when $\lambda = 1$ in Eq.11, i.e., only local factors are used for graph classification, the performance drops significantly. This shows that global latent variables carry critical graph level information in the generative process. We remark that these global/local representations are learned in unsupervised settings; then the representations are tested in SVM classifiers.

## I   HOW DOES NODE LEVEL TASK PERFORMANCE GETS EFFECTED WITH GLOBAL LEVEL INFORMATION?

As a detailed look at the Table 2, we indicate in detail how the node level performance gets effected with all the values of $\eta$ in Figure 6 and Table 6 for the Citeseer dataset. We like to elaborate that the input to the decoder during unsupervised model training ($\hat{\mathbf{z}}_l(i) = \eta \, \mathbf{z}_l(i) + (1 - \eta) \, \mathbf{z}_g$) is the same which goes to the SVM classifier during the inference and downstream task performance evaluation. Therefore, although we show the downstream task performance here, it has a strong relatability with the GL-Disen decoding process. We could see that when $\eta = 0$, when only global level disentangled information is used for decoding process, GL-Disen is unable to learn local patch level disentangled information. Therefore during inference of GL-Disen and evaluation of downstream node classification, the performance is very low. This is expected as all nodes in the graph will have the same features, as the global latent representation is common to all the nodes. But when disentangled local information start to get incorporated in the decoding process with the increase of

$\eta$ value, we can observe the increase of downstream task performance. But after $\eta = 0.45$, again the performance has started to degrade upto $\eta = 1$ (when GL-Disen does not learn global graph level information). This shows that to obtain the best performance, both global and local disentangled information is crucial for node level tasks and GL-Disen provides the capability of controlling the contribution of each factor to obtain the optimal performance.

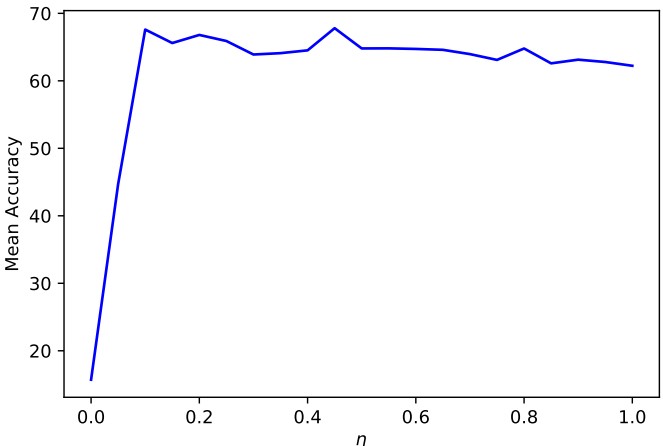

Figure 6: Impact on Mean test set accuracy on Node classification with different $\eta$ on CiteSeer.

Table 6: Mean test set accuracy change on Node classification with different $\eta$ on CiteSeer.

| $\eta$ | MEAN ACCURACY % |
|---|---|
| 0.0 | 15.7 |
| 0.05 | 44.7 |
| 0.1 | 67.6 |
| 0.15 | 65.6 |
| 0.2 | 66.8 |
| 0.25 | 65.9 |
| 0.3 | 63.9 |
| 0.35 | 64.1 |
| 0.4 | 64.5 |
| **0.45** | **67.8** |
| 0.5 | 64.8 |
| 0.55 | 64.8 |
| 0.6 | 64.7 |
| 0.65 | 64.6 |
| 0.7 | 64.0 |
| 0.75 | 63.1 |
| 0.8 | 64.8 |
| 0.85 | 62.6 |
| 0.9 | 63.1 |
| 0.95 | 62.8 |
| 1.0 | 62.2 |

## J  ABLATION STUDY

### J.1  IMPACT OF THE RATIO BETWEEN THE NUMBER OF FACTORS USED FOR GLOBAL AND LOCAL LATENT REPRESENTATIONS

In this study we analyse the impact of the number of factors we allocate for storing global latent $\mathbf{z}_g$ and local latent $\mathbf{z}_l(i)$ has on the downstream task. We consider the number of factors for each latent as the dimension of the vectors each of them are kept in the GL-Disen. Therefore when changing the number of factors allocated for each latent, we change the vector size. This is done as a ratio. For an example, if the ratio is 3:1, that means the dimensionality of $\mathbf{z}_g$ is three times the dimensionality of $\mathbf{z}_l(i)$. When we input for the decoder also, we maintain this ratio.

| Global to Local Ratio | Accuracy $\pm$ S.D. % |
|:---:|:---:|
| 3:1 | $86.5 \pm 0.8$ |
| 1:1 | $90.7 \pm 0.6$ |
| 1:3 | $88.7 \pm 0.6$ |
| 0:1 | $87.7 \pm 0.7$ |

Above table shows the impact of the change of factor sizes for global and local latents have on MUTAG dataset's classification accuracy. When both latents have same size, the size was 128 in this trained model. We can observe that the accuracy drops when most of the information comes from global latent as the reconstruction error increases by being unable to reconstruct the graph in the decoder. When there is no dimension for global, that is the vanilla GVAE where all information is entangled and considered local. This shows that allowing equal amount of factors is the beneficial for our GL-Disen.

### J.2  IMPACT OF USING DEGREE FEATURES

Social network datasets (IMDB, REDDIT) we are using do not have input node features. Therefore existing work has taken two approaches to provide synthetic node features. First is using a constant vector and the second is to use node's degree count as a feature. Following table shows the performances of GL-Disen when different types of synthetic features are used.

| Synthetic feature | IMDB-BINARY | IMDB-MULTI | REDDIT-BINARY | REDDIT-MULTI-5K |
|:---:|:---:|:---:|:---:|:---:|
| Constant | $72.5 \pm 0.3$ | $48.3 \pm 0.3$ | $90.9 \pm 0.3$ | $54.9 \pm 0.1$ |
| Out degree count | $74.7 \pm 0.6$ | $52.1 \pm 0.4$ | $85.5 \pm 0.4$ | $52.4 \pm 0.2$ |

We can observe that, only IMDB benefits from degree features while Reddit achieves best performance when constant feature is used.

### J.3  IMPACT OF HIDDEN SIZE

Following table shows how hidden size effect the performance of GL-Disen.

| Hidden size | MUTAG | PTC-MR | IMDB-BIN | IMDB-MUL | REDD-BIN | RED-MUL-5K |
|:---:|:---:|:---:|:---:|:---:|:---:|:---:|
| 32 | $90.5 \pm 0.3$ | $63.7 \pm 1.3$ | $71.9 \pm 0.6$ | $49.9 \pm 0.1$ | $90.9 \pm 0.3$ | $54.9 \pm 0.1$ |
| 128 | $90.7 \pm 0.6$ | $65.9 \pm 1.2$ | $72.7 \pm 0.4$ | $51.7 \pm 0.5$ | $90.2 \pm 0.2$ | $52.4 \pm 0.2$ |
| 512 | $89.8 \pm 0.8$ | $67.9 \pm 0.8$ | $74.7 \pm 0.6$ | $52.1 \pm 0.4$ | $89.3 \pm 0.2$ | $51.6 \pm 0.2$ |

### J.4  SELECTED FINAL SET OF HYPER-PARAMETERS

Following has the final set of hyper-parameters we selected for reported results. We observed that reconstructing only node features was beneficial for MUTAG and PTC as they had node features. For other datasets, we only reconstructed adjacency matrices.

| Hidden size | MUTAG | PTC-MR | IMDB-BIN | IMDB-MULTI | RED-BIN | RED-MULTI-5K |
|---|---|---|---|---|---|---|
| $\alpha$ | 1e-6 | 1e-9 | 1e+3 | 10 | 1e+4 | 1e+4 |
| $\gamma$ | 1e-6 | 1e-9 | 1e+3 | 10 | 1e+4 | 1e+4 |
| C | 10 | 10 | 1e+4 | 1 | 1e+3 | 1e+3 |
| # epochs | 20 | 30 | 40 | 30 | 50 | 50 |
| Hidden size | 128 | 512 | 512 | 512 | 32 | 32 |
| Degree | No | No | Yes | Yes | No | No |
| Reconstruction | Node | Node | Adj | Adj | Adj | Adj |

# K   MORE SAMPLE PLOTS FOR DISENTANGLEMENT ANALYSIS

## K.1   SAMPLES FOR DISENTANGLEMENT BETWEEN GLOBAL AND LOCAL LATENT FACTORS SAMPLES FROM LEARNT POSTERIOR DISTRIBUTIONS BY GL-DISEN

Figure 7 provides more examples to showcase the ability of our proposed GL-Disen model's capability of producing mutually exclusive information for its global and local latents. This shows that GL-Disen actually can disentangle two latent factors.

## K.2   SAMPLES FOR SHOWING THE SIMILARITY OF GLOBAL LATENT FACTORS ACROSS THE GRAPH COMPARED TO LOCAL LATENT FACTORS

In this section, we provide more qualitative proof that the latent factors our GL-Disen model maps as global are indeed globally common across the graph. Figure 8 and Figure 9 compares the pairwise similarity of global latent factors and local latent factors separately in two matrices for multiple different graphs from MUTAG dataset. We can see that compared to local level inter-node latent representation similarity, global level similarity is very high as the Mean Absolute pairwise difference is low.

# L   GENERATED DENSER GRAPHS FROM GL-DISEN FOR SYNTHETIC DATASET

In the main paper we have only incorporated graph generated with very small probabilities as dense graphs are hard to visualize. In Figure 10 we have incorporated denser graphs.

# M   DETAILED COMPARISON OF GL-DISEN WITH SOME RELATED WORK

The graph-graph proximity (Bai et al., 2019) is not a disentanglement learning based method. It has proposed an unsupervised learning mechanism where they first obtain graph embedding using multihead attention to aggregate nodes in each layer and concatenate K layers together. The unsupervised loss they are using is based on graph proximity there they have to calculate the proximity measure for each and every pair of graphs in the dataset to obtain the distance matrix( Fig 1.(b) of (Bai et al., 2019)). Compared to this graph pair-wise comparison, GL-Disen uses a disentanglement mechanism individually for each graph where it forces the model to make the global level information and local level information in the graph to be independent. This implicitly pushes the model towards learning similar global representations of graphs in the dataset. As shown in Fig. 4 of our updated manuscript, even without this expensive pairwise comparison, global level similarities has been captured by our GL-Disen. In Fig. 4(b) we show that our global representation ($\mathbf{z}_g$) has been able to map different global generative factor values (global generative factor is the feature that similarize/group graphs in this dataset) to different latent values. Hence during inference it has been able to apply it back and increase the edge density of any random graph. Since they have randomly split the dataset for train, validation and test splits while we conduct cross validation, we are unable to compare the performance.

Apart from FactorGCN (Yang et al., 2020) being a supervised model (section 3.4) while GL-Disen is unsupervised, there are two main differences we have with them. First is that, although they disentangle the generative factors, they do not determine whether those are local or global. In our work our intention is to factorize the graph into two separate levels of information which are,

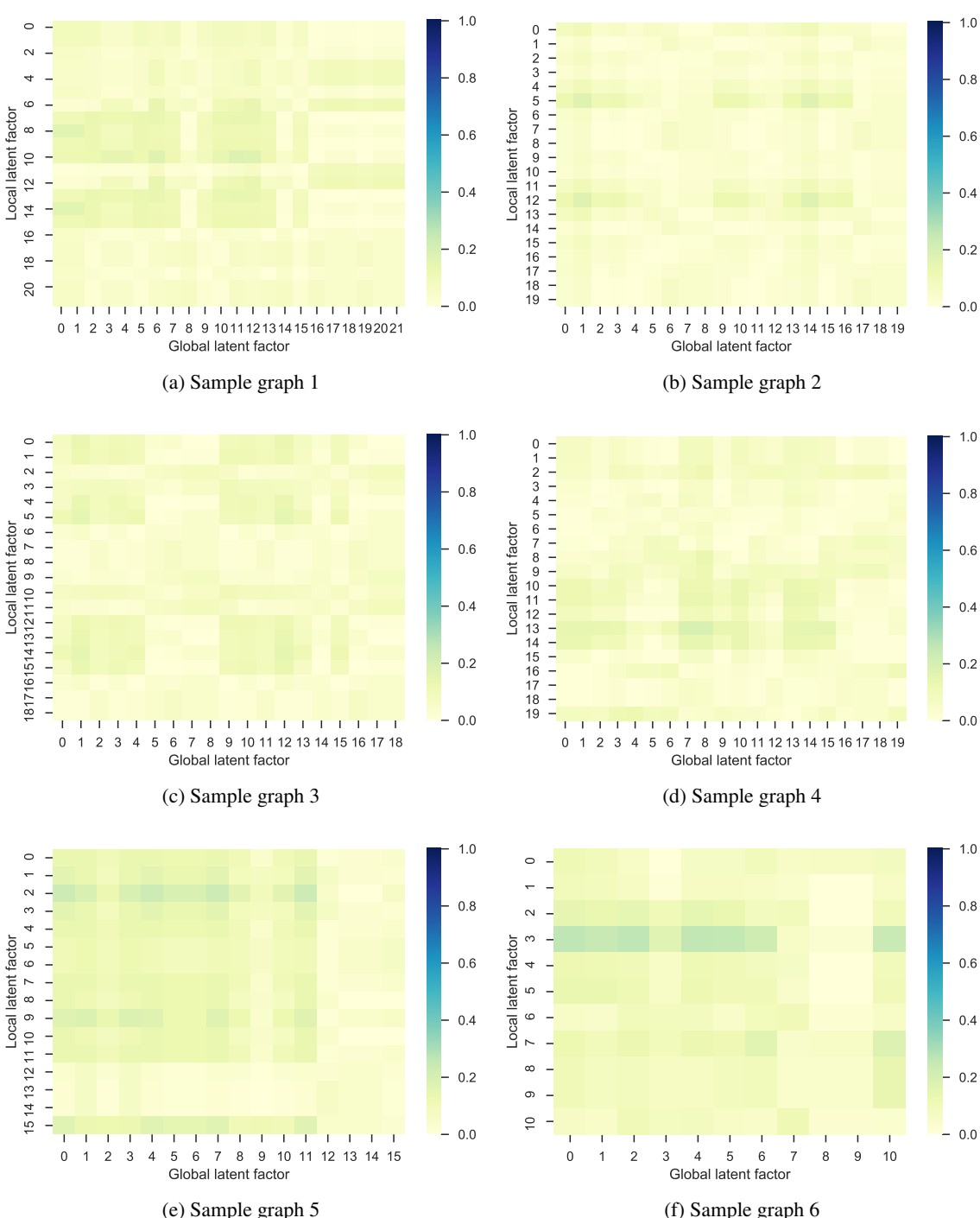

Figure 7: Correlation matrices to show the dependence among global and local latent factors. Very low correlation close to 0.0 shows that these is very less correlation between these two factors. These sample graphs from MUTAG dataset showcase the capability of GL-Disen in producing mutually exclusive information for its global and local latent factors.

information common to the entire graph and information specific to local patches. FactorGCN has no mechanism to ensure any extracted factor is global. Second difference is they disentangle the input features before the neighbourhood propagation while we disentangle after the propagation.

This is due to the difference of objectives we have with FactorGCN. We want to extract global level information common for the entire graph, the more suitable design decision for us is to disentangle after neighbourhood propagation allowing the graph to share information with its neighbours and form local patches. This mechanism also helps us when some datasets like Reddit and IMDB which do not have any node features to extract more meaningful information as our method does not get guidance from any supervision as well. We are unable to compare directly with results provided in FactorGCN paper as it is a supervised method and ours is unsupervised.

NED-VAE (Guo et al., 2020) also does not disentangle factors common for the entire graph (global factors) from the factors specific for each local patch. Their unsupervised disentangle mechanism aims at disentangling node features, edge features and node-edge joint. Using the loss function term A (Sec 4.2.2 (Guo et al., 2020)) they try to make node, edge and node-edge joint features independent of each-other. We do not impose such restrictions in our model as we only want to separate out features common for the entire graph and features specific to local patches. Our model has the flexibility of using either node or edge or joint features and extract globally relevant information from any of these, while also separating out local features which are specific for patches. Their node-edge joint representation is like a combination of both nodes and edges . In Fig 3 (Guo et al., 2020), third column samples, it seems like while node and edge factors have disentangled graph features (first two columns of Fig 3), node-edge joint factor has entangled them again (Both edge density and node values change with it). We believe this is expected, as it is mainly used to inform node and edge decoders about the structure of that particular individual graph (since node and edge encoders have no other mechanism to share information among them). Basically it captures the uniqueness of each individual graph with respect to node and edge features and structures, while our global latent tries to capture the common global generative factors which the entire graph dataset was generated from. For GL-Disen, we demonstrate in Appendix G that removing individual node/edge/patch specific local information using disentanglement and separating out global factors is beneficial for graph level tasks such as classification. This work has only focused on graph generation. Hence discriminative performance on downstream tasks have not being assessed in the paper.

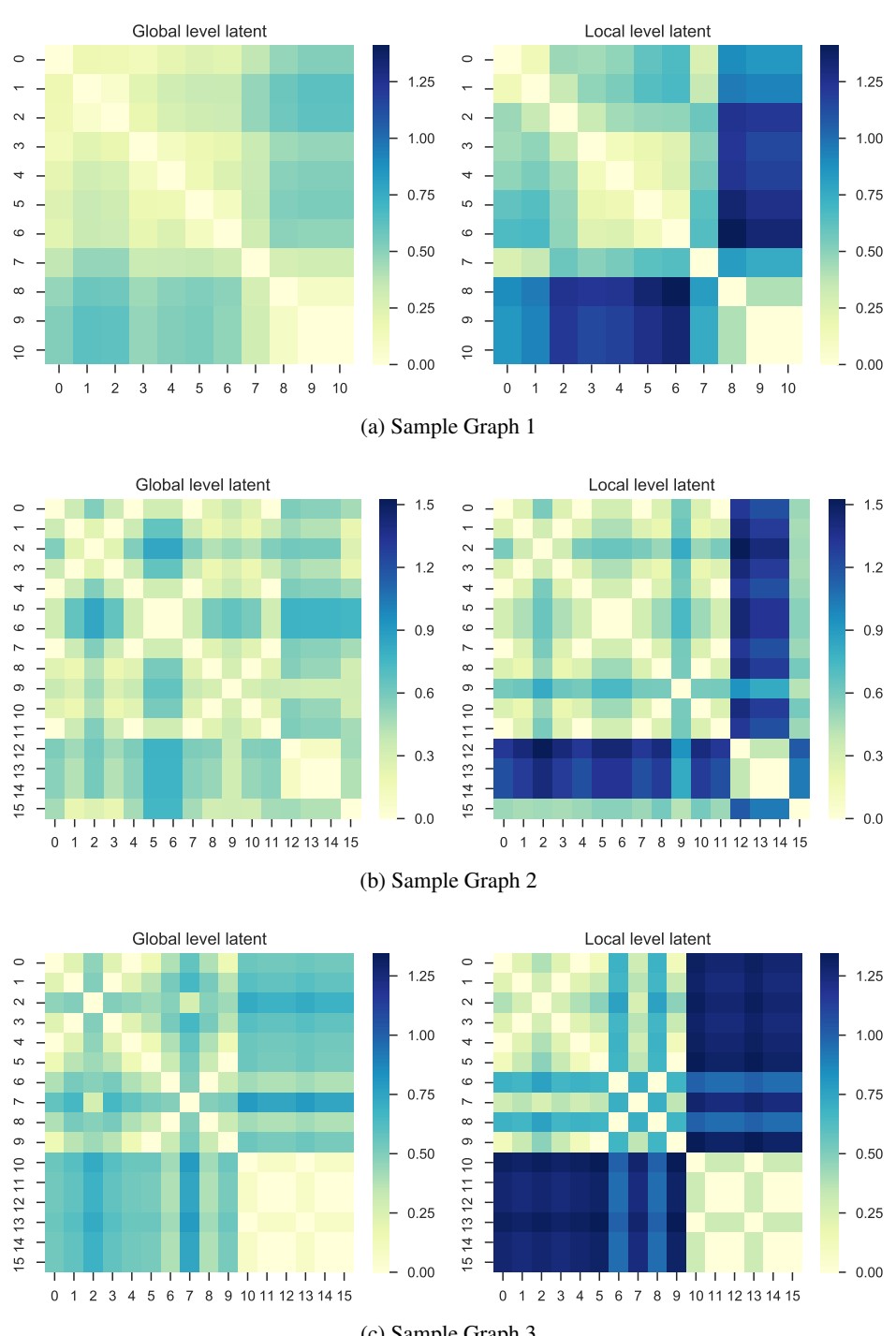

Figure 8: Part 1 : Mean Absolute Pairwise Difference among global latent factors and local latent factors for two graphs from MUTAG. These matrices clearly show that the inter-node latent representation difference for produced global latent factors is very low compared to local latent factors. This indicates the global factor representations disentangled by GL-Disen is indeed shared among the entire graph, unlike local factors which are local to certain nodes groups.

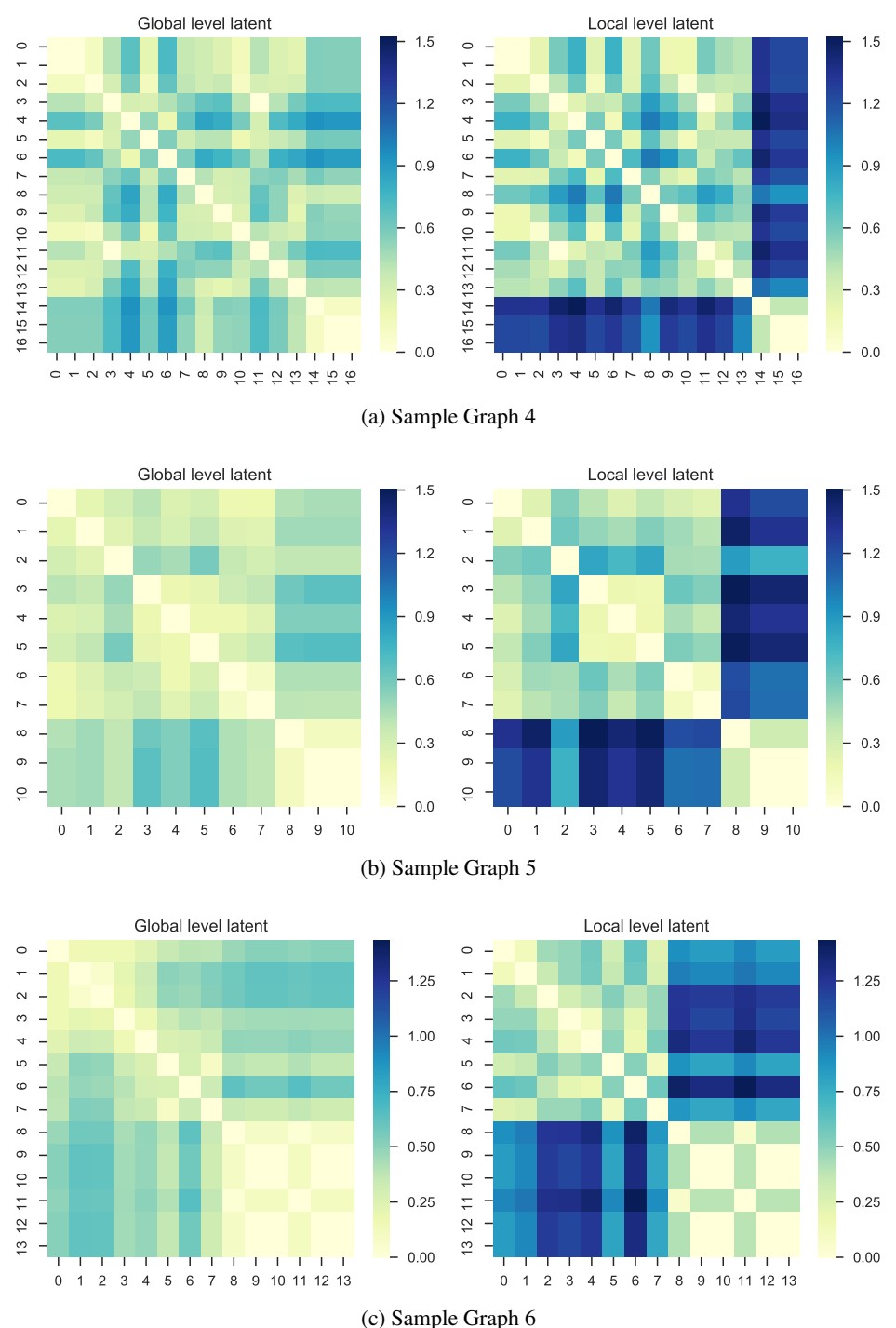

(a) Sample Graph 4

(b) Sample Graph 5

(c) Sample Graph 6

Figure 9: Part 2 : Mean Absolute Pairwise Difference among global latent factors and local latent factors for two graphs from MUTAG. These matrices clearly show that the inter-node latent representation difference for produced global latent factors is very low compared to local latent factors. This indicates the global factor representations disentangled by GL-Disen is indeed shared among the entire graph, unlike local factors which are local to certain nodes groups.

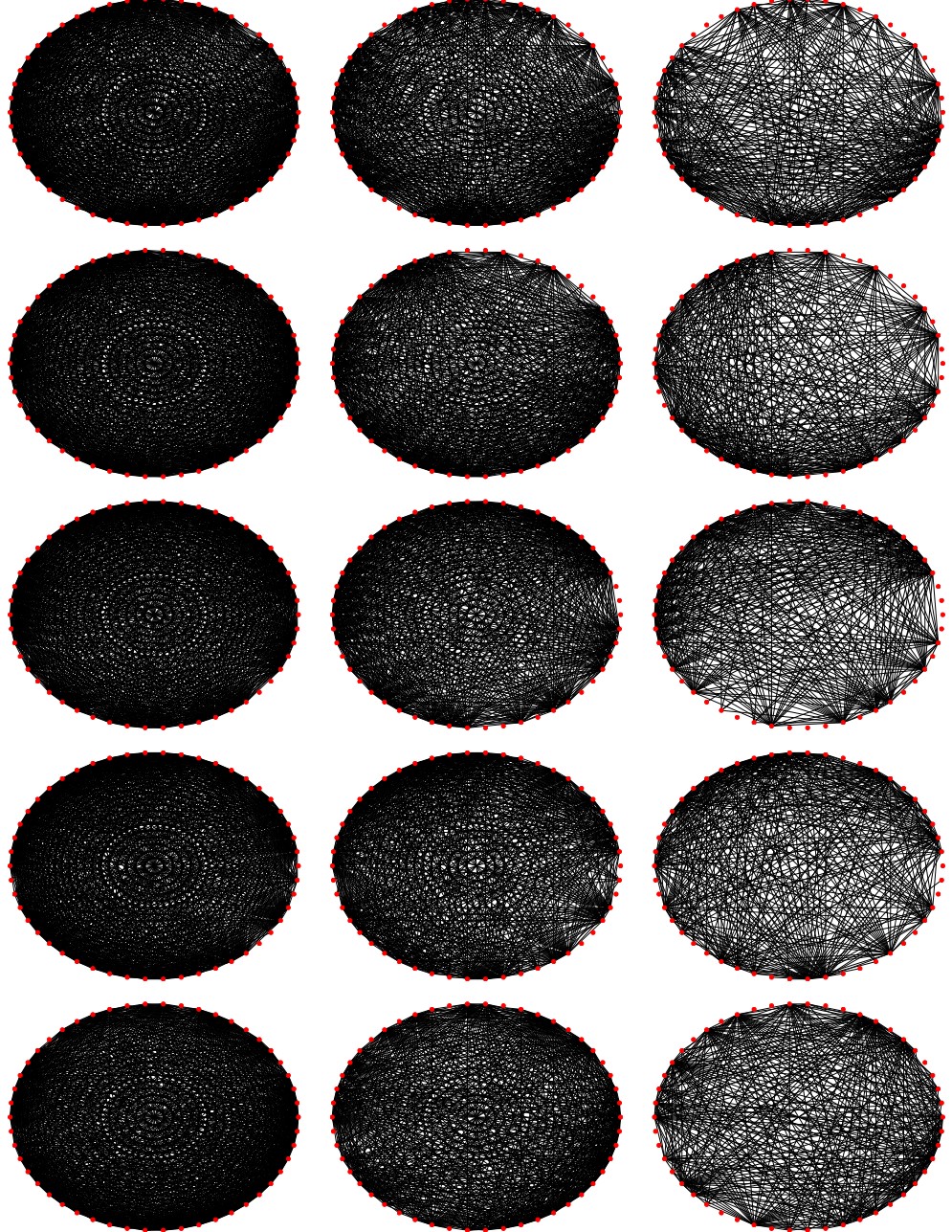

Figure 10: Visualizes denser generated graphs where in each row local latent representation $\mathbf{z}_g$ is fixed and in each column $\mathbf{z}_g$ is fixed. This shows that $\mathbf{z}_g$ has a strong negative correlation with the global generative factor.

