# OpenReview forum: "GL-Disen: Global-Local disentanglement for unsupervised learning of graph-level representations"
_ICLR.cc/2021/Conference — Reject_

### Official Review · AnonReviewer3 · 2020-10-28
**The authors proposed a disentanglement learning based approach for unsupervised graph level representation learning, which aim to capture the global and local latent factors.**

**Rating:** 5
**Confidence:** 3

**Review:**

In this paper, the authors proposed a disentanglement learning based approach for unsupervised graph level representation learning. They assume that disentangled representations which capture these global and local generative factors into independent latent units can be highly beneficial for graph level tasks. The extensive experiments and analysis  show that our method achieves the state-of-the-art performance on the task of unsupervised graph representation learning.

===========
Strengths:
1. The paper is well written and the disentangling factors can benefit the unsupervised graph representation learning.
2. The performance of this work is good compared with the state-of-the-art baselines. The source code is also available.
3. The related work is sufficient to understand the motivation of this work. The

=====
Weakness:
1. The idea is not very novel. For example, two important assumptions 1) a  global and local factor for graph analysis 2) local latent factors are independent.
Those two assumptions actually have been explored in unsupervised learning tasks. For example, the follow paper[1] exactly disentangle local and global information into two separate sets of latent variables within the VAE framework.  It seems that migrating this idea under graph is straightforward. The paper is more like a mixture of [1] and (Higgins et al., 2017), and GCN

[1] Charakorn, Rujikorn, et al. "An Explicit Local and Global Representation Disentanglement Framework with Applications in Deep Clustering and Unsupervised Object Detection." arXiv preprint arXiv:2001.08957 (2020).

2. In Figure4, It seems that the GL-Disen global has very good accuracy. The GL-disen global-local combines only outperform GL-Disen global within a very small range of \lambda but with large fluctuation. Does that mean the local factor contribution little to the overall performance?


In Conclude,the authors propose a VAE based learning algorithm to disentangle the global graph-level information. The overall presentation is good. The similar ideas have been explored in unsupervised learning. The novelty of this work is thus not very impressive.

---

> ### Author Response · Authors · 2020-11-23
> **Response to AnonReviewer3**
>
> We thank the reviewer for the useful feedback.
>
> **Q1**: “the follow paper[1] exactly disentangle local and global information into two separate sets of latent variables within the VAE framework. It seems that migrating this idea under graph is straightforward.”
> [1] Charakorn, Rujikorn, et al. "An Explicit Local and Global Representation …”
>
> We thank reviewer for pointing out [1]. We read it very carefully. **We believe [1] cannot be applied to our problem** (if we overlook some details, we hope to receive reviewer’s feedback and we deeply appreciate reviewer’s comment).
>
> [1] has an interesting and simple idea. The key idea is to produce auxiliary data and to pass that into another VAE (Fig.2). In their approach, **producing aux data is crucial and not easy** ([1], Sec 3): aux data needs to be created such that it contains only local information, excluding global information. Even for images, they need to do that meticulously, choosing different patch sizes manually for different datasets (Fig 3), so that they can (i) preserve local correlations between pixels within each patch (ii) reduce global long-range correlations between pixels (sec 3.1). [1] has done that **manually** for different datasets (sec 4.2.2, last paragraph): “larger patch sizes can result in z_l occasionally being used to represent the digit identity” (thus (ii) cannot be satisfied); “for CelebA, by visual inspection, we find that a patch size too small can worsen the disentanglement” ((i) cannot be satisfied).
>
> In our problem, we do not want to perform such meticulous tuning and supervision. In fact, we follow the problem setup in [Higgins et al. 2017]: Unsupervised disentangled learning of complex data where no a priori knowledge of the generative factors exists, and little to no supervision for discovering the factors is available. (our apology if this is unclear)
>
> Even we change our problem setup and allow meticulous manual supervision, graphs are significantly much more difficult than images. For example, if we consider datasets like MUTAG or PTC which contains graphs representing properties like heteromatic nitro and carcinogenicity, a person needs thorough knowledge of the domain to understand what these means and what graph structures are meaningful and what are not. Therefore, it is very difficult to produce aux data in order to apply the methods in [1]: what transformation can be used to retain local corrections and reduce global long-range correlations? How could we know it is a good transformation so that we can produce the aux data and pass it as input for [1]? For images, human can view the data and perform tuning ([1], Fig 3); For graphs, it is not easy for human to assess, specially some graphs are very specialized such as those in MUTAG or PTC.
>
> **This discussion highlights some subtle but important details of global / local disentanglement for graphs, which our work is the first to study.** In our work, we add an accumulation step on top of Beta-VAE to force the emergence of global information (our Fig 1). This accumulation step allows us to disentangle global / local information in unsupervised settings, extract graph level representation from the global information, and we do not need to understand the complex meanings of the specialized graph datasets. Our only assumption is that graph level generative factors produce common effects to all vertices of the same graph.
>
> Please see our response to AnonReviewer5  “Note about novelty”  regarding the difference between our work and Beta-VAE.
>
> ---
>
> **Q2**. “In Figure 4 … Does that mean the local factor contribution little to the overall performance?”
>
> Yes, reviewer is correct and this is our message. Fig 4 (Fig 5 of updated manuscript) shows graph classification accuracy - a global task. With global/local disentanglement, local latent factors capture only local variations which are distractions to a global task. Therefore, not including them and using only global representation (red line) achieves the best performance.
>
> We thank the reviewer for comments, especially careful review including Appendix.

---

> > ### Author Response · Authors · 2020-11-24
> > **Response to AnonReviewer3 - Further clarification**
> >
> > We thank again the reviewer for the useful feedback and pointing out [1].
> >
> > We would like to highlight that in our revised paper we have included new discussion regarding the difference between our work and  Beta-VAE (Higgins et al., 2017). Please find the discussion at the end of Section 3.
> >
> > We hope our response has clarified reviewer's concern especially related to [1] Charakorn, Rujikorn, et al.
> >
> > We would be grateful if reviewer can reassess our work in light of these clarifications, and we deeply appreciate reviewer’s valuable time.

---

### Official Review · AnonReviewer1 · 2020-10-28
**The paper proposed GL-Disen, which is a disentanglement-based unsupervised method for graph representation learning.**

**Rating:** 6
**Confidence:** 4

**Review:**

In this paper, the authors proposed to disentangle the global level information from the local level one to reduce the effect of the irrelevant information. The proposed method outperforms several state-of-the-arts on multiple datasets for graph classification. Overall, I like the idea of applying unsupervised disentangled learning to graph level representation learning. Some concerns are on the experimental study and missing references.

Strong Points:
1. Disentanglement learning is a cutting-edge field and has gained much attention in recent years. It is true that global and local features often entangle together when we learn graph representations. The problem is real and important.

2. The architecture of the model is easy to understand and reasonable.

3. The experimental study is comprehensive, including both qualitative analysis and quantitative analysis. The experimental setup instructions and pseudo-codes are very clear, making the algorithm easy to be reproduced.

Weak Points:
1. Performing experiments only on graph classification tasks weakens the significance of the paper. It is common for graph representation learning methods to be tested on other tasks, such as graph similarity/distance computation and graph-level clustering, in order to draw a general and convincing conclusion.

2. Some important references are missing. The authors should discuss and compare with them.
On graph-level representation learning:
- Bai et al. Unsupervised Inductive Graph-Level Representation Learning via Graph-Graph Proximity. IJCAI 2019.
On disentangled representation learning:
- Yang et al. Factorizable Graph Convolutional Networks. NIPS 2020.
- Guo et al. Interpretable Deep Graph Generation with Node-edge Co-disentanglement. KDD 2020.

3. The paper mentioned that the global and local latent generative factors are sampled from their respective posterior distributions. More details are expected.

---

> ### Author Response · Authors · 2020-11-24
> **(Continued) Response to AnonReviewer1 - Part 1**
>
> We thank the reviewer for the useful feedback
>
> **Comment 1 - Evaluation Tasks**
>
> In our paper, one main reason for selecting graph classification is as follows. Currently, infomax principle based methods dominate unsupervised graph-level representation learning, and results are reported for the graph classification task (InfoGraph[1], CMV[2]). Therefore, we focus on this task so that we can compare to their results properly, to understand how our proposed global-local disentanglement approach compares with the infomax approach for unsupervised graph-level representation learning.
>
> While state-of-the-art infomax methods like InfoGraph[1], CMV[2] aggregate all information from all the patches to generate the global representation, our proposed GL-Disen has an explicit specialized mechanism to remove irrelevant local information for graph level representations (without aggregating all); that is disentangling. We wanted to evaluate performance of this approach: explicit removal of irrelevant local information and retain of global information, and how this approach compares with methods which have no explicit mechanisms to remove irrelevant local information, for graph level tasks.
>
> In addition, in the updated manuscript Sec 4.3, we have extended our experiments for node level tasks as well. As the focus of our paper is on disentangling global graph-level latent representations and local patch-level representations, we conducted focused experiments to show how these disentangled information affects node level tasks and graph level tasks. For graph level tasks, we show superior performance when using only global representation, and this validates our hypothesis that local information is distractive for graph-level tasks. For node-level tasks, interestingly, we show better performance when local representation is combined with some global information ($z_g$). This observation is consistent with recent work GraphWave[3] which has stated that identifying distant nodes with similar neighbourhood structures is a strong fact for node level task performance. Our combined method increases the performance due to the fact that $z_g$ has been able to capture those long distance similarities. Although recent methods such as DGI [4] have claimed that their “derived patch representations are driven to preserve mutual information with the global graph summary, this allows for discovering and preserving similarities on the patch-level” for long distance, they neither have empirical analysis nor capability to explicitly evaluate the optimal amount of global information required. In our work, our global/local disentangled representations from GL-Disen enable us to explicitly control the amount of global/local information and combine them, and we are able to explicitly demonstrate that combination of global/local can achieve the best performance for node level task, please see our Table 6 (Appendix I) of our experiments.
>
> We also added Sec 4.2.2 to the updated manuscript to elaborate more on the explanations of our disentangled factors.
>
> [1] InfoGraph: Unsupervised and Semi-supervised Graph-Level Representation, ICLR 2020
> [2] Contrastive Multi-View Representation Learning on Graphs, ICML 2020
> [3] Learning Structural Node Embeddings via Diffusion Wavelets, KDD 2018
> [4] Deep Graph Infomax, ICLR 2019

---

> > ### Author Response · Authors · 2020-11-24
> > **Response to AnonReviewer1 - Part 2**
> >
> > **Comment 2 - Comparison with existing disentangling methods**
> >
> > Thank you for pointing out these references. We have carefully gone through these works. Detailed comparison can be found in Appendix M and we have updated our related work section to include these.  Overall, our idea of global-local disentanglement is distinctive to these works.
> >
> > Although the graph-graph proximity [Bai et al. IJCAI’19] is aimed at unsupervised learning, their approach is different and they did not propose any ideas related to disentanglement. The other major difference this method has with GL-Disen and all other latest graph learning models we have compared is that, this uses a pairwise graph comparison mechanism to learn graph level similarities  (Bai et al. IJCAI’19 Fig 1.(b)). Our proposed GL-Disen does not require this expensive comparison to learn good graph representations, as our method removes irrelevant local information via disentanglement.
> >
> > FactorGCN [Yang et al. NIPS 20] does not have any mechanism to learn global level factors which are common to the entire graph. Although it can learn a set of factors under which nodes are connected to each-other, they cannot determine which factors are locally important and which are globally relevant. On the other hand, FactorGCN is a supervised model (sec 3.4 FactorGCN [Yang et al. NIPS 20]) while our work GL-Disen is unsupervised.
> >
> > NED-VAE [Guo et al.KDD 20] also does not disentangle factors common for the entire graph (global factors) from the factors specific for each local patch. Their unsupervised disentangle mechanism aims at disentangling node features, edge features and node-edge joint. Using the loss function term A (Sec 4.2.2 [Guo et al.KDD 20]) they try to make node, edge and node-edge joint features independent of each-other. We do not impose such restrictions in our model as we only want to separate out features common for the entire graph and features specific to local patches. Our model has the flexibility of using either node or edge or joint features and extract globally relevant information from any of these, while also separating out local features which are specific for patches. Their node-edge joint representation is like a combination of both nodes and edges . In Fig 3 [Guo et al.KDD 20], third column samples, it seems like while node and edge factors have disentangled graph features (first 2 columns of Fig 3), node-edge joint factor has entangled them again (Both edge density and node values change with it). We believe this is expected, as it is mainly used to inform node and edge decoders about the structure of that particular individual graph (since node and edge encoders  have no other mechanism to share information among them). Basically it captures the uniqueness of each individual graph with respect to node and edge features and structures, while our global latent captures the common global generative factors which the entire graph dataset was generated from. For GL-Disen, we demonstrate in Appendix G that removing individual node/edge/patch specific local information using disentanglement and separating out global factors is beneficial for graph level tasks such as classification.
> >
> > ---
> > **Comment 3** “ The paper mentioned that the global and local latent generative factors are sampled from their respective posterior distributions. More details are expected.”
> >
> > The outputs from the encoder of our GL-Disen are two sets of parameters for each patch in the input graph. First set is the mean and variance values of the individual distributions of local latent factors. Each patch in the graph has its own posterior distribution. Then we sample a local latent representation for each patch from these individual posterior distributions. Since each patch/node can be different from one another, sampling from individual distributions separately is reasonable. The second set of mean and variance parameters output from GL-Disen encoder are, to represent a posterior distribution which is common for all the patches of the graph: the distribution for global latent factors. Unlike patch-wise individual distributions for local latent, global latent has only a single distribution for the entire graph (We include an accumulation operation in Eq. 9 of the paper to obtain a single posterior distribution). Then we sample a single global latent variable from this distribution as the global latent representation which is used for graph level tasks.
> >
> > ---
> > We are trying to conduct additional experiments for other tasks as suggested by the reviewer.

---

### Official Review · AnonReviewer4 · 2020-10-28
**The paper tries to study unsupervised disentanglement learning for graph-level representations. In particular, it focuses on the complex process with global and local generative factors and proposes a VAE based learning algorithm, which argues achieving state-of-the-art performance on the task of unsupervised graph representation learning.**

**Rating:** 4
**Confidence:** 4

**Review:**

I think the idea of the paper is interesting. The writing is well and easy to read. However, it does not meet the condition of acceptance from my point of view. I have some concerns with its characterization of the literature.

- Some important related work is missing. It seems authors ignore talking about some literature of unsupervised graph representation learning, such as [1], [2], etc. Also, they do not make a performance comparison with the methods above in experiments.
[1] Contrastive Multi-View Representation Learning on Graphs. ICML 2020
[2] Self-supervised Training of Graph Convolutional Networks. Arxiv 2020

- Disentangling the global and local generative factors graph representation learning is important. However, the authors didn't explain the definition of “Global” and “Local” factors clearly. It would also be better if they can show an example of global/local factors when generating graph.

- The experiments are missing. I have some concerns as follows. What is the best number of generative factors which is important for this method? Can this method occur mode collapse and how to valid or prevent it? How can this method prove that each factor is necessary for the generative process? What is the real meaning of each factor? How about the time/space complexity of this method? More experiments or discussions should be conducted to answer these questions.

Based on the above reasons, this paper can have much more improvement.

---

> ### Author Response · Authors · 2020-11-24
> **(Continued) Response to AnonReviewer4 - Part 1**
>
> We thank the reviewer for the useful feedback.
>
> **Point 1**
> Thank you for pointing out these related works [1], [2]. Sorry for unclear, **but in our original submission, we have already compared GL-Disen with Contrastive multi-view [1] in Table 1 (CMV) and discussed it in our Section 2 - Related work.** Following the same experiment setup, our proposed method outperforms CMV consistently across all datasets.
> Regarding [2] self supervised training on GCN paper, they propose self-supervised strategies such as link removing and feature covering to improve GCN’s feature learning ability. In particular, **[2] does not propose any mechanism for graph level representation learning** as GCN is already known for its patch level feature learning ability. The focus of our work is graph level representation using a new global/local disentanglement approach.
> [1]Contrastive Multi-View Representation Learning on Graphs, ICML 2020
> [2]Self-supervised Training of Graph Convolutional Networks. Arxiv 2020
>
> ---
>
> **Point 2**
> Apologies if we did not explain the definition of global and local factors clearly. The definitions of global/local are as we mentioned in the abstract: “We propose a VAE based learning algorithm to
> disentangle the global graph-level information, which is common across the entire
> graph, and local patch-level information, which varies across individual patches
> (the local subgraphs centered around the nodes).” We have some explanation in the introduction, but we agree more explanation is better.
>
> In the revised manuscript, following the reviewer’s suggestion, we have added new synthetic graph analysis for better explanation of global/local, pls see Section 4.2.2. The synthetic graphs are Erdos-Renyi ER graphs. The ER(n,p) graphs are synthetic graphs with two global generative factors: number of nodes n and a parameter p \in [0, 1] for the synthetic graph to include an edge (i, j) for 1 <= i < j <=n with probability p. In our experiment, we focus on parameter p, as n is too easy to learn. Therefore, we create a training dataset of ER(n,p) with fixed n and varying p. In this dataset, the global generative factor is p and the local factor is local randomness. We verify that our method can discover global / local factors by performing traversal of the learned latent variables, similar to other disentanglement learning work such as Beta-VAE [a], InfoGAN [b] (but they focused on images). We emphasize that our setup is **unsupervised**: the generative factor p is unknown to our method, and our method discovers this global factor.
>
> For real-world data, a priori knowledge of global/local factors is usually not available, e.g. molecular graphs, which are very specialized. For graphs modeling a Reddit discussion thread, one of the global factors could be the topic of their comments, because all the users have discussed about it. But it is very difficult to know the exact set of global/local factors, similar to other real-world graph data.
>
> It should be noted that our method does not require a priori knowledge of global/local factors. Our method disentangles the factors into global and local, captures the global factors into graph representation, and this is sufficient for many graph level tasks such as classification. There is no need for us to understand the semantics of these global factors.
>
> [a]beta-VAE: Learning Basic Visual Concepts with a Constrained Variational Framework, ICLR 2017
> [b] InfoGAN: Interpretable Representation Learning by Information Maximizing Generative Adversarial Nets, NIPS 2016

---

> > ### Author Response · Authors · 2020-11-24
> > **(Continued) Response to AnonReviewer4 - Part 2**
> >
> > **Point 3** Answers for each question:
> >
> > **Q1: What is the best number of generative factors which is important for this method?**
> > Empirically, we have performed experiments on the effects of different latent variable dimensions (Appendix J.3, was Appendix E.3 in our original submission).
> >
> > Analytically, however, it is very difficult to derive the best number of generative factors. For many complex data such as molecular graphs, a priori knowledge of the underlying generative factors is not available. Also, in our unsupervised setup, we do not get any supervision from downstream tasks during training.
> >
> > Our main focus of GL-Disen is to separate out the group of generative factors as local and global, i.e. global/local disentanglement. Then we extract graph level representations from the set of global factors. **Importantly, there is no need for us to explicitly separate each one of global and local factors from those sets individually.**
> >
> > **Q2: Can this method occur mode collapse, how to valid or prevent it?**
> > The scope of this work is on evaluating how disentanglement can be equipped for learning graph level representations and we proposed a GVAE based approach as a proof of concept. Indeed posterior collapse is a fundamental problem for VAE. Our method builds on top of GVAE, therefore our work may suffer a certain amount of posterior collapse. There are fundamental works to address posterior collapse [1,2]. In principle we can integrate such ideas into our system to alleviate mode collapse and this may further improve the performance.
> > [1] Preventing posterior collapse with  δ-VAES, ICLR’19
> > [2] Avoiding Latent Variable Collapse with Generative Skip Models, AISTATS’19
> >
> > **Q3: How can this method prove that each factor is necessary for the generative process?**
> > In what follows, we discuss how we validate that our learned local and global latent variables carry critical information for the generative process.
> >
> > As we discussed, our main focus of GL-Disen is to separate out the group of generative factors as local and global, i.e. global/local disentanglement, and that is sufficient for our task. There is no need for us to explicitly separate each one of global and local factors from those sets individually.
> >
> > To evaluate the necessity of our local/global latent variables, first we calculated the node feature reconstruction error for MUTAG dataset and obtained the following results. MSE when both global and local factors fed to the decoder is 0.03256 and it increases to 0.03654 when global factors are removed (local only). When only global factors are fed to the decoder (global only), the error further increases to 0.08329. From these errors, we can observe that local latent factors have the largest impact on the generation of the node features. This is expected as global factors are common for all the patches for a given graph. Therefore, to reconstruct the node features (which differ from node to node), local factors are crucial. However, we can observe from the difference of full model and local only errors, that our model does not ignore the global factors during node feature generation. Hence showing it is also necessary.
> >
> > Next, we show that our learned global latent variables carry critical graph level information in the generative process. We refer to sec 4.2.2 - Synthetic graph based experiments and Fig. 4 on the updated manuscript. In Fig 4(b), we show generated sample graphs using disentangled global and local factors. In each row of Fig 4(b), the local latent factors are fixed and in each column the global factors are fixed. When we consider a single row, we could observe that, the edge density of the graph changes with the change of global factors. Although 2 rows have two structurally different graphs (nodes have different neighbourhoods), the global factor has been able to change the edge density of those 2 in a similar manner. If only local factors are necessary, then every graph in the same row should look alike. This shows that graph level generative information is captured by global latent variables. Therefore the global latent variables are necessary for the generative process.
> >
> > Further evidence that our learned global latent variables carry critical graph level information comes from the evaluation on graph classification task. In Appendix G, we evaluate the impact of different combinations of global/local latent (Eq. 11) on graph level task performance. We observe that using only learned global latent variables ($\lambda$ = 0) achieves the best performance in graph level classification. On the other hand, when $\lambda$ = 1 in Eq. 11, i.e., only local factors are used for graph classification, the performance drops significantly. This shows that global latent variables carry critical graph level information in the generative process. We remark that these global/local representations are learned in unsupervised settings; then the representations are tested in SVM classifiers.

---

> > > ### Author Response · Authors · 2020-11-24
> > > **Response to AnonReviewer4 - Part 3**
> > >
> > > **Q4: What is the real meaning of each factor?**
> > >
> > > In our analysis discussed in sec 4.2.2 - Synthetic graph based experiments, we show that learned global latent variable captures the graph-level generative factor of the Erdos-Renyi graphs, i.e., probability p for the synthetic graph to include an edge between node i and node j. Furthermore, the learned local latent captures the local randomness in the generative process of Erdos-Renyi graphs.
> > >
> > > However, for many real-world complex data/graphs, meanings of generative factors are not available (Higgins et al. (2017)), e.g. global/local generative factors for molecular graphs are not accessible except for molecular scientists. On the other hand, our method requires no a priori knowledge of these factors. In particular, even though there is no knowledge regarding the meaning of the global / local factors, our method can capture global factors into a representation, excluding the local factors. This representation is sufficient as our main focus is on graph level tasks.
> > >
> > >
> > > **Q5: Complexity Analysis**
> > >
> > > We like to discuss the time and space complexity of GL-Disen compared to our baseline GVAE. Most of the computation complexity comes from the GNN encoder (Eq.8) where the time and space complexity is $O(V^2)$ for a single GNN layer with $V$ number of nodes in the graph and for GNN with $N$ layers, it becomes $O(V^2N)$. Only difference between GL-Disen encoder and GVAE encoder is that due to disentangling, and GL-Disen encoder requires output of two different parameter sets for global and local posterior distributions instead of one as in baseline GVAE. Therefore we need an additional 2 GNN layers. Since it is a constant addition, the overall complexity stays at $O(V^2N)$ scale. The decoder complexity for both GVAE and GL-Disen is $O(V^2)$ with adjacency reconstruction being the dominant component (Eq.10). The two additional steps GL-Disen have for the disentanglement are as follows (in between encoder and decoder): (A) Accumulating using Eq.9 and (B) combining global and local samples to feed to the decoder. Both step (A) and (B) are linear operations during both training and inference with the complexity of $O(V)$ in both time and space. Compared to the high complexity of the GNN encoder and decoder common for both GVAE and ours, this linear increment to disentanglement is not significant.
> > > In the updated manuscript, we have included this time/space complexity analysis of our method.

---

### Official Review · AnonReviewer2 · 2020-10-30
**Lack of guarantee that the global and local factors are disentangled, unclear definition, limited novelty**

**Rating:** 3
**Confidence:** 4

**Review:**

The authors propose a VAE-type generative model approach to characterize the hidden factors, with a divided focus on the global and local reconstructions. The claim is that the learnt hidden representations are disentangled (which is not defined clearly) using two reconstruction terms. The setting of the problem adopts the graph VAE setting in [1,2] (which I think the authors should mention in the related work), and the ELBO & local aggregation (convolution) approaches used in this paper are relatively standard in the generative modelling and graph representation learning domain.

Apart from the limited novelty, which would not have affected my evaluation if it solves the problem as claimed, I have several major concerns about this paper:

1. The notion of disentanglement is not well-defined in the first place. In the VAE setting where the hidden factors are stochastic, does disentanglement refer to independence? Or they are orthogonal under a specific measure induced by the graph itself? The claims made by the authors can never be examined rigorously (the visual results do not constitute supportive evidence as I shall discuss later).

2. There is no guarantee that the so-called global and local factors are not confounded. Both the global and local reconstruction terms involve the two types of factors. Given the high expressivity of deep learning models, the local factors can easily manage both tasks, or the global factors are merely enhancing the signals of the local factors. There no mechanism to prevent the cross-terms during the optimization, so the learning process of the global and local factors confounded as a result of how the authors design the objective function.

3. Unclear interpretation of the visual results. It seems that the visual results showcase a similar pattern among the local and global factors, despite the difference that the signal is stronger for the local factors (which is evident as they play a more critical role in the objective). In the absence of a clear definition of disentanglement, more persuasive numerical results and interpretations are needed.


[1] Kipf T N, Welling M. Variational graph auto-encoders[J]. arXiv preprint arXiv:1611.07308, 2016.
[2] Xu, Da, et al. "Generative graph convolutional network for growing graphs." ICASSP 2019-2019 IEEE International Conference on Acoustics, Speech and Signal Processing (ICASSP). IEEE, 2019.

---

> ### Author Response · Authors · 2020-11-24
> **(Continued) Response to AnonReviewer2 - Part 1**
>
> We thank the reviewer for the useful feedback. We have updated our manuscript to clarify the unclear areas you have mentioned and discuss in detail here.
>
>
> ### **Clarifications about the novelty of this work**
>
>
> Regarding reviewer's comment “The setting of the problem adopts the graph VAE setting in [1,2] (which I think the authors should mention in the related work), and the ELBO & local aggregation (convolution) approaches used in this paper are relatively standard in the generative modelling and graph representation learning domain.”
>
> Thank you for pointing out that we have overlooked the citation in places where we have referred GVAE in the paper. We fixed that.
>
> Our apologies if our writing is not clear, but the main contribution of our work is global/local disentanglement for graph representation learning. **The techniques mentioned by the reviewer: graph VAE (Kipf and Welling), ELBO, convolution are our backbone system and basic components. Critically, these techniques mentioned by the reviewer (i.e. graph VAE, ELBO, convolution) are not sufficient to achieve global/local disentanglement for graphs.** Therefore, our main contribution is a new method built on top of these techniques mentioned by the reviewer **and other critical ideas to achieve global/local disentanglement for graphs, as explained below.**
>
> We have updated the manuscript and we hope this can clear any misunderstanding (end of Sec 3). In particular, our method is built on top of Beta-VAE [Higgins et al. 2017], which has been applied mostly to images. However, there is a key difference between Beta-VAE and our work. As reviewer may know, **Beta-VAE baseline cannot discover global factors automatically:** Beta-VAE discovers independent latent factors, but there is no way for a baseline Beta-VAE to understand if these factors are global / non-global in the unsupervised setting. Usually, some manual inspection is performed on the learned latent variables. E.g., for images, one needs to perform traversal of individual latent variables one by one, and observe their effects (e.g. change in azimuth). Please see Beta-VAE Figure 2. Not only this would require manual effort but also this could be very difficult for graphs, since many graphs represent very specialized knowledge, e.g. protein-protein interaction, and it is very difficult to understand the observed effects and determine which factors are global/local.
>
> In our work, we add on top of Beta-VAE **an accumulation step** for the GNN encoder outputs of vertices belonging to the same graph (see Fig 1 in our paper). This forces **automatic** emergence of the global factors - common information across all the vertices. This mechanism is critical for our idea to extract representation for the whole graph, and we are able to capture global factors without a priori knowledge of the generative factors.
>
> **Regarding another paper [2], they did not propose any idea related to disentangling,** and it is about a variation of GVAE for growing graphs. While they propose a variation of GVAE for growing graphs where the generation of each new node is conditioned on all existing nodes, **we aim at a variation of GVAE which is capable of disentangling global and local factors from a fixed graph. [2] does not have a mechanism for capturing global latent information** as their adaptive ELBO in eq 7 [2] term 2 is node wise unlike our global KL divergence single term for the entire graph in our ELBO (2nd term eq 6 of our paper). Due to this, our work is significantly different from [2].
>
>
> [2] Xu, Da, et al. "Generative graph convolutional network for growing graphs." ICASSP 2019-2019 IEEE International Conference on Acoustics, Speech and Signal Processing (ICASSP). IEEE, 2019

---

> > ### Author Response · Authors · 2020-11-24
> > **(Continued) Response to AnonReviewer2 - Part 2**
> >
> > **Answers for the major concerns**
> >
> > **Comment 1** “The notion of disentanglement is not well-defined in the first place. In the VAE setting where the hidden factors are stochastic, does disentanglement refer to independence? Or they are orthogonal under a specific measure induced by the graph itself? The claims made by the authors can never be examined rigorously (the visual results do not constitute supportive evidence as I shall discuss later).”
> >
> > Our apology if our notion of disentanglement is not clear, **but our notion of disentanglement is the common one widely used in existing work [a,b]**. Following [a, b], a disentangled representation can be defined as one where a single latent unit is sensitive to changes in single type of generative factors, while being relatively invariant to changes in other types of factors.
> >
> > **We humbly point out that there are differences between disentanglement and independence, as explained in [b].**
> >
> > **Our methods in examining disentanglement are the same as existing work in disentangle learning.** Measuring correlation among disentangled latent factors (Fig 2(a) of our updated paper) is a commonly used mechanism to evaluate the quality of disentanglement [c,d]. Also calculating pair-wise embedding difference using Mean Absolute Pairwise Difference was first proposed in [b] which we used (Fig 2(b) of our updated paper) to compare the amount of patch wise variation for each latent factor to showcase that MAPD for global factors is low as it is common to all the patches. (In the last answer we experimentally show that both local and global factors lie on the same value range and the signal magnitudes are similar; the reason for high MAPD for local is due to its high variation across patches.)
> >
> > In addition, we have added more analysis in Sec 4.2.2 in our updated paper where we used a synthetic dataset with a known global generative factor and demonstrate that our global latent factor ($z_g$) is the one which maps to this global generative factor, not the local factor. We also show generated graphs to visualize the impact of our disentangled factors (both global and local) on the GL-Disen’s generative process to verify that our model indeed disentangles those two factors. We remark that such visualization is common in disentanglement learning work to validate disentanglement (most previous work focused on images) [b, e].
> >
> > ---
> >
> > **Comment 2** “There is no guarantee that the so-called global and local factors are not confounded. Both the global and local reconstruction terms involve the two types of factors. Given the high expressivity of deep learning models, the local factors can easily manage both tasks, or the global factors are merely enhancing the signals of the local factors. There no mechanism to prevent the cross-terms during the optimization, so the learning process of the global and local factors confounded as a result of how the authors design the objective function.”
> >
> >
> > Our apology if this is unclear, but as we have clarified above, **we do have a proposed mechanism of accumulation on top of Beta-VAE to disentangle global and local factors (see Figure 1 in our paper), and we have adopted analysis methods in existing disentangling works [b,c,d] to validate that our learned representations are indeed disentangled to a large extent as we discussed in Comment 1. Specifically, in Sec. 4.2.1 in our paper we demonstrate the correlation between our global and local factors are very close to 0.0 showing they are able to capture different variations/generative factors in input data which don’t correlate.**
> >
> > In addition, To elaborate  that our learned global latent variables carry critical graph level information in the generative process, we refer to sec 4.2.2 - Synthetic graph based experiments and Fig. 4 on the updated manuscript. In Fig 4(b), we show generated sample graphs using disentangled global and local factors. In each row of Fig 4(b), the local latent factors are fixed and in each column the global factors are fixed. When we consider a single row, we could observe that, the edge density of the graph changes with the change of global factors. Although 2 rows have two structurally different graphs (nodes have different neighbourhoods, which was captured by local factors), the global factor has been able to change the edge density of those 2 in a similar manner. If only local factors are necessary and global factors merely enhance signals of local factors, then every graph in the same row should look alike. This shows that graph level generative information is captured by global latent variables. Therefore the global latent variables are necessary for the generative process.

---

> > > ### Author Response · Authors · 2020-11-24
> > > **Response to AnonReviewer2 - Part 3**
> > >
> > > **Comment 2** answer continued ...
> > >
> > > Further evidence that our learned global latent variables carry critical graph level information comes from the evaluation on graph classification task. In Appendix G, we evaluate the impact of different combinations of global/local latent (Eq. 11) on graph level task performance. We observe that using only learned global latent variables ($\lambda$ = 0) achieves the best performance in graph level classification. On the other hand, when $\lambda$ = 1 in Eq. 11, i.e., only local factors are used for graph classification, the performance drops significantly. This shows that global latent variables carry critical graph level information in the generative process. We remark that these global/local representations are learned in unsupervised settings; then the representations are tested in SVM classifiers.
> > >
> > >
> > > ---
> > >
> > > **Comment 3** “It seems that the visual results showcase a similar pattern among the local and global factors, despite the difference that the signal is stronger for the local factors”
> > >
> > > To validate that local embedding values are not larger in magnitude (stronger signals) than their global counterparts, here we plot the histograms (Please click on the dropbox link [View Histogram](https://www.dropbox.com/s/nqyuqsub8r8l4wo/mutag_plots_histo_6.pdf?dl=0) as inline images are not supported here.) of both local and global latent patch embeddings for the graph in Fig 2(b) of updated manuscript. Histogram in the left illustrates how global factor representations (embeddings) range which has a value range of (0.2 - 0.4). On the right we illustrate the local latent representation value range of 0.0 - 0.4. From this we can clearly observe that both global factor embeddings and local factor embeddings are in the same value range. Local factors don’t have stronger values. The only difference is that local factors have larger value variation among patches.
> > >
> > > **In conclusion, reviewer’s statement “despite the difference that the signal is stronger for the local factors” is not correct. We remark that the numerical results and interpretations in our analysis are widely used in existing disentanglement learning works [b,c,d].**
> > >
> > > [a] Representation learning: A review and new perspectives. In IEEE Transactions on Pattern Analysis & Machine Intelligence, 2013
> > > [b]beta-VAE: Learning Basic Visual Concepts with a Constrained Variational Framework, ICLR 2017
> > > [c] Disentangled graph convolutional networks, ICML, 2019
> > > [d] Unsupervised model selection for variational disentangled representation learning. ICLR 2020
> > > [e] InfoGAN: Interpretable Representation Learning by Information Maximizing Generative Adversarial Nets, NIPS 2016.
> > >
> > > ---
> > >
> > > Again, our apologies if there was anything unclear in our paper which may have caused misunderstanding. We hope our discussion above and our updated paper can clarify the misunderstanding. We humbly request the reviewer to reassess our work in light of these clarifications, and we deeply appreciate reviewer’s valuable time.

---

### Official Review · AnonReviewer5 · 2020-11-06
**An interesting idea, but it is unclear whether the improvement really comes from disentanglement**

**Rating:** 5
**Confidence:** 4

**Review:**

Summary:
This paper proposes an unsupervised graph-level representation learning method considering global-local disentanglement. Specifically, the authors propose a GL-Disen model based on graph VAE architecture to jointly learn global and local representations for a graph. The global information is shared across the whole graph while the local information varies from patch to patch, corresponding to common and local factors, respectively. Empirical experimental results show that the learned representation achieves superior performance in the downstream graph classification task, and analyses demonstrate the learned representations exhibit some disentangle property.

Pros:
1. Unsupervised graph representation learning considering global and local disentanglement seems to be a novel problem.
2. The proposed method generalizes disentangled VAE into graph data to disentangle common factors from the local ones. The formulations and model descriptions are clear.
3. Experiments, including both qualitative analysis and quantitative results, demonstrate the effectiveness of the learned global factors in downstream tasks.

Cons and questions:
My major concern lies in the insufficiency of experiments. Specifically:
1. The disentanglement part is modified from Beta-VAE. Since normal VAE is adopted in graphs (e.g., Variational Graph Auto-Encoders by Kipf and Welling), the authors need to compare these methods to demonstrate the improvement is actually from the disentanglement part rather than the VAE structure.
2. Although the authors demonstrate the effectiveness of disentanglement in downstream tasks (i.e., graph classification), it is unclear whether these global factors have intuitive explanations on some of the datasets, e.g., the showcases of molecular graphs in Duvenaud et al., 2015, or the authors may adopt some synthetic datasets.
3. Since both the global and local node representations are disentangled, I am curious whether the local node representations can also be validated in some downstream node-level tasks.
4. Figure 2 in Section 4.2.1 is not entirely convincing since there is no reference line of how much correlation a non-disentangled method will have (e.g., in Ma et al., 2019, the authors compare the disentangled method with GCN).

Other questions:
5. How the proposed method can handle the mode collapse problem, i.e., only a few latent factors learn useful information?
6. As shown in Table 1, though the proposed method outperforms other GNNs, it does not always compare favorably to kernel-based methods such as GCKN. The authors may want to further elaborate on the pros and cons of using GNN vs. kernel-based methods.
7. There lacks a discussion on the complexity of the proposed method.
8. The technical contribution is somewhat limited since both Beta-VAE and graph VAE are known in the literature. It would be more interesting if the authors can integrate local-global disentanglement with local neighborhood disentanglement in Ma et al. 2019 to derive a more novel architecture.

I will be happy to improve my scores if authors can address the above questions.


=========

I have updated my score considering the paper has improved its quality after the revision (adding more experiments/baselines, comparison with the literature, etc.).


=========
New updates: following the new comments of Reviewer 4, I also briefly check the code in the supplementary material and find it indeed seems to have the consistency problem (i.e., not reconstructing graph edges as mentioned in the paper). Thus, I am also wondering how the authors implement Graph-VAE in the rebuttal phase and whether the improvement of their proposed method over Graph-VAE is really from disentanglement or the differences in the autoencoder. Based on this potentially serious problem, I reinstate my original score and think the paper should be clarified before acceptance.

---

> ### Author Response · Authors · 2020-11-23
> **(Continued) Response to AnonReviewer5 - Part 1**
>
> We thank the reviewer for the constructive feedback.
>
> **Note about novelty:**
>
> To our best knowledge, global/local disentanglement for graph-level representation learning is novel, as reviewer mentions.
>
> Reviewer is correct that our work is based on Beta-VAE and graph VAE which are known in the literature. But we would like to highlight one key difference which is critical for this whole work (our apologies, our writing is not clear to highlight this). As reviewer knows, **Beta-VAE baseline cannot discover global factors automatically:** Beta-VAE discovers independent latent factors, but there is no way for a baseline Beta-VAE to understand if these factors are global / non-global in the unsupervised setting. Usually, some manual inspection is performed on the learned latent variables. E.g., for images, one needs to perform traversal of individual latent variables one by one, and observe their effects such as change in azimuth (Beta-VAE, Figure 2). Not only this would require manual effort but also this could be very difficult for graphs, since many graphs represent very specialized knowledge, e.g. protein-protein interaction, and it is difficult to understand what factors are global/local.
>
> In our work, we add on top of Beta-VAE **an accumulation step** for the GNN encoder outputs of vertices belonging to the same graph (see Fig 1 in our paper). This forces the emergence of the global factors - common information across all the vertices. This mechanism is critical for our idea to extract representation for the whole graph, and we are able to capture global factors without a priori knowledge of the generative factors.
>
> We have updated the manuscript to make this clear (end of Section 3).
>
> ---
>
> **Answers for the Major concerns:**
> 1.  We updated our main experiment results in Table 1 with GVAE baseline (Kipf and Welling) for all the datasets and showed consistent and considerable improvement we obtained via the disentanglement mechanism in our GL-Disen. Note that, without disentanglement, baseline GVAE performs poorer than recent work InfoGraph [a] in most cases, while with the addition of disentanglement our GL-Disen outperforms InfoGraph consistently. GVAE baseline and GL-Disen use the same backbone networks and training parameters for fair comparison.
>
> 2. In order to discuss the intuitive meaning of what global latent variables capture, as reviewer suggested, we added analysis with a synthetic dataset in Section 4.2.2. Specifically, we perform experiments on Erdos-Renyi ER graphs. The ER(n,p) graphs are synthetic graphs with two global generative factors: number of nodes n and a parameter p \in [0, 1] for the synthetic graph to include an edge (i, j) for 1 <= i < j <=n with probability p. In our experiment, we focus on parameter p, as n is too easy to learn. Therefore, we create a training dataset of ER(n,p) with fixed n and varying p. We pass this training dataset to our GL-Disen method. We demonstrate that GL-Disen can discover the generative factor p using the training dataset only. In particular, we follow previous work such as Beta-VAE [b], InfoGAN [c] to perform traversal of the latent variables. We show that our representation is disentangled: our global latent variable ($z_g$) captures the global generative factor p, and local latent variable captures local randomness. We emphasize that our setup is unsupervised: the generative factor p is unknown to our method, and our method discovers this global factor.
>
> 3. To evaluate the impact of GL-Disen has on node level, we added section 4.3 for node classification tasks. We observe that for graph level tasks, completely removing local information achieves the best accuracy (Appendix G), and this validates our hypothesis that local information distracts graph level tasks. For node level tasks, we found out some combining of local and global information achieves the best performance. This is consistent with the observation in recent work  GraphWave[e] which has stated that identifying distant nodes with similar neighbourhood structures is a strong fact for node level task performance. Our combined method increases the performance due to the fact that $z_g$ has been able to capture those long distance similarities. Although methods like DGI [d] have claimed that their “ derived patch representations are driven to preserve mutual information with the global graph summary, this allows for discovering and preserving similarities on the patch-level” for long distance, they neither have empirical analysis nor  capability to determine the optimal amount of global information required.  In contrast, GL-Disen disentangles local information and global information apart. These disentangled representations allow us to explicitly control the amount of global/local information and combine them, please see our Table 6 (Appendix I) of our experiments.

---

> > ### Author Response · Authors · 2020-11-23
> > **Response to AnonReviewer5 - Part 2**
> >
> > 4. We updated Figure 2 (a) to compare the disentanglement results by GL-Disen with our baseline GVAE which does not have any disentangling capability of global and local level information. The main difference between GL-Disen and GVAE is that GL-Disen has an accumulating module to enable global/local disentanglement. The network architectures and training parameters are the same. We observed that correlation of global and local latent variables for GL-Disen is almost 0.0, while that for GVAE is considerably higher around 0.4. This is because GVAE neither has an explicit mechanism to determine global and local information nor anyway to achieve global/local disentanglement.
> > ---
> >
> > **Answers for other questions:**
> >
> > 5. The scope of this work is on evaluating how disentanglement can be equipped for learning graph level representations. Indeed posterior collapse is a fundamental problem for VAE. Our method builds on top of GVAE, therefore our work may suffer a certain amount of posterior collapse. There are fundamental works to address posterior collapse [1,2]. In principle we can integrate such ideas into our system to alleviate mode collapse and this may further improve the performance.
> > [1]  Avoiding Latent Variable Collapse with Generative Skip Models  AISTATS 2019
> > [2]  Preventing posterior collapse with  δ-VAES, ICLR 2019
> >
> > 6. One of the major aspect of kernel methods is they use manual processes (graph traversals like depth first search) to find all possible paths for substructures like random walks, trees or graphlets. Then they compare all those pairs of paths in each pair of graph to calculate kernel values to find similarities. This is a very expensive operation. However for small graphs this gives better results as it covers all possible neighbourhoods. However as the GCKN [f] paper mentions, when there are very large dense graphs, they are unable to extend this method. This can be a reason that kernel based methods do not evaluate on denser datasets like Reddit. On the other hand, GNNs achieve efficiency by eliminating from manual path and graph to graph pairwise comparison and reducing neighbourhoods for only random walks. However even with limited neighbourhood, we could see that GNNs specially with our disentanglement mechanism have been able to achieve almost similar performance.
> >
> > 7. We like to discuss the time and space complexity of GL-Disen compared to our baseline GVAE. Most of the computation complexity comes from the GNN encoder (Eq.8) where the time and space complexity is $O(V^2)$ for a single GNN layer with $V$ number of nodes in the graph and for GNN with $N$ layers, it becomes $O(V^2 N)$. Only difference between GL-Disen encoder and GVAE encoder is that due to disentangling, and GL-Disen encoder requires output of two different parameter sets for global and local posterior distributions instead of one as in baseline GVAE. Therefore we need an additional 2 GNN layers. Since it is a constant addition, the overall complexity stays at $O(V^2 N)$ scale. The decoder complexity for both GVAE and GL-Disen is $O(V^2)$ with adjacency reconstruction being the dominant component (Eq.10). The two additional steps GL-Disen have for the disentanglement are as follows (in between encoder and decoder): (A) Accumulating using Eq.9 and (B) combining global and local samples to feed to the decoder. Both step (A) and (B) are linear operations during both training and inference with the complexity of $O(V)$ in both time and space. Compared to the high complexity of the GNN encoder and decoder common for both GVAE and ours, this linear increment to disentanglement is not significant.
> >
> > 8. At the beginning of this response, we discuss the difference between our method and Beta-VAE, which enables discovery of global factors in the unsupervised setting.
> >
> > [a] InfoGraph: Unsupervised and Semi-supervised Graph-Level Representation, ICLR 2020
> > [b] beta-VAE: Learning Basic Visual Concepts with a Constrained Variational Framework, ICLR 2017
> > [c] InfoGAN: Interpretable Representation Learning by Information Maximizing Generative Adversarial Nets, NIPS 2016
> > [d] Deep Graph Infomax, ICLR 2019
> > [e] Learning Structural Node Embeddings via Diffusion Wavelets, KDD 2018
> > [f] Convolutional Kernel Networks for Graph-Structured Data, ICML 2020

---

> > > ### Comment · AnonReviewer5 · 2020-11-24
> > > **Thanks for the responses**
> > >
> > > I thank the authors for their detailed responses and for revising the paper substantially. I believe the paper has improved its quality after the revision and thus improve my score accordingly.
> > >
> > > A quick follow-up comment: though I appreciate the synthetic dataset added in the revision, estimating the parameter p in the ER graph may be too simple since it can be estimated directly by the number of edges / the number of nodes^2. Adopting more complicated synthetic datasets, e.g., the stochastic block model or the forest fire model, will make the results more convincing.

---

> > > > ### Author Response · Authors · 2020-11-25
> > > > **Thank you**
> > > >
> > > > We thank the reviewer for your valuable time and constructive feedback which helped us to improve our work.
> > > > Thank you for your response and suggestion.

---

### Author Response · Authors · 2020-11-24
**Rebuttal Paper Revision**

We thank all the reviewers for their valuable feedbacks.

Updated manuscript contains analysis, experiments and explanations requested by our reviewers.
All additions and updates to the original submission are indicated in blue for easy reference.

**Updates to the main paper**
1. End of Sec. 3 - Comparison with $\beta$-VAE
2. Fig 2 - (a) Added the GVAE reference of entangled method for correlation comparison
3. Sec. 4.2.1 - Added the GVAE reference of entangled method for correlation comparison
4. Sec 4.2.2 - Added a Synthetic dataset and experiments to elaborate intuitive meaning of disentangled factors
5. Section 4.3 - Node level task evaluation

**Updates to the Appendix**
1. Appendix C - Model complexity Analysis
2. Appendix F - Moved the comparison of GL-Disen with Kernel methods to here and discussion of pros and cons of Kernel vs GNN
3. Appendix H - How can this method prove that each factor is necessary for the generative process?
4. Appendix I - Detailed analysis of Node level task
5. Appendix A - added more detailed discussion comparing existing disentangled methods to ours

---

### Decision · Program_Chairs · 2021-01-07
**Final Decision**

**Decision:**

Reject

**Comment:**

In this paper, the authors designed a disentanglement mechanism for global and local information of graphs and proposed a graph representation method based on it. I agree with the authors that 1) considering the global and local information of graphs jointly is reasonable and helpful (as shown in the experiments) and 2) disentanglement is different from independence.

However, the concerns of the reviewers are reasonable --- Eq. (2) and the paragraph before it indeed show that the authors treat the global and the local information independently. Moreover, the disentanglement of the global information (the whole graph) and the local information (the patch/sub-graph) is not well-defined. In my opinion, for the MNIST digits, the angle and the thickness (or something else) of strokes can be disentangled (not independent) factors that have influences on different properties of the data. In this work, if my understanding is correct, the global and the local factors just provide different views to analyze the same graphs and the proposed method actually designs a new way to leverage multi-view information. It is not sure whether the views are disentangled and whether the improvements are from "disentanglement".

If the authors can provide an example to explain their "disentanglement" simply as the MNIST case does, this work will be more convincing. Otherwise, this work suffers from the risk of overclaiming.